# Establishment of AIRS Climate-Level Radiometric Stability using Radiance Anomaly Retrievals of Minor Gases and SST

L. Larrabee Strow[1] and Sergio DeSouza-Machado[1]

[1]Department of Physics and JCET, University of Maryland Baltimore County, Baltimore, Maryland

**Correspondence:** L. Larrabee Strow (strow@umbc.edu)

**Abstract.** Temperature, $H_2O$, and $O_3$ profiles, as well as $CO_2$, $N_2O$, $CH_4$, CFC12, and SST scalar anomalies are computed using a clear subset of AIRS observations over ocean for the first 16-years of NASA's EOS-AQUA AIRS operation. The AIRS Level-1c radiances are averaged over 16 days and 40 equal-area zonal bins and then converted to brightness temperature anomalies. Geophysical anomalies are retrieved from the brightness temperature anomalies using a relatively standard optimal estimation approach. The $CO_2$, $N_2O$, $CH_4$, and CFC12 anomalies are derived by applying a vertically uniform multiplicative shift to each gas in order to obtain an estimate for the gas mixing ratio. The minor gas anomalies are compared to the NOAA ESRL in-situ values and used to estimate the radiometric stability of the AIRS radiances. Similarly the retrieved SST anomalies are compared to the SST values used in the ERA-Interim reanalysis and to NOAA's OISST SST product. These inter-comparisons strongly suggest that many AIRS channels are stable to better than 0.02 to 0.03 K/Decade, well below climate trend levels, indicating that the AIRS blackbody is not drifting. However, detailed examination of the anomaly retrieval residuals (observed minus computed) show various small unphysical shifts that correspond to AIRS hardware events (shutdowns, etc.). Some examples are given highlighting how the AIRS radiances stability could be improved, especially for channels sensitive to $N_2O$ and $CH_4$. The AIRS short wave channels exhibit larger drifts that make them unsuitable for climate trending, and they are avoided in this work. The AIRS Level 2 surface temperature retrievals only use short wave channels. We summarize how these short wave drifts impacts recently published comparisons of AIRS surface temperature trends to other surface climatologies.

## 1 Introduction

The Atmospheric Infrared Sounder (AIRS) on NASA's AQUA satellite platform (Aumann et al., 2003) measures 2378 high-spectral resolution infrared radiances between 650 and 2665 cm$^{-1}$ with a resolving power ($\lambda/\Delta\lambda$) of ~1200. Launched in 2002 into a sun-synchronous polar orbit with a 13:30 ascending node equator crossing time, AIRS now has been operating almost continuously for 17+ years.

The long record of AIRS allows measurements of short-term climate trends that are especially useful given it's global coverage. Nominal decadal climate temperature trends are in the 0.1-0.2K/decade range. For example a recent Intergovernmental Panel on Climate Change (IPCC) report (Masson-Delmotte et al., 2018) suggests 20th century trends (2000-2017) of about 0.17K/decade. If AIRS is to contribute to climate-level trend measurements, uncertainty estimates for the time stability of the

AIRS radiances are a pre-requisite before using AIRS Level 2/3 products for climate level trending. Estimating the level of any instrument-related trends, for a wide range of AIRS channels, is the subject of this work.

A recent study (Aumann et al., 2019) addressed the stability of a single AIRS channel by comparisons to sea surface temperatures (SST). Some limitations of that study are addressed below, but its major limitation is that is evaluates only one channel. AIRS retrievals use 400+ AIRS channels, and there is no guarantee that the AIRS stability in one channel applies to all channels, as acknowledged in (Aumann et al., 2019).

AIRS is sensitive to a host of atmospheric and surface variables, including atmospheric temperature (via $CO_2$ emissions), humidity, surface temperature, $O_3$, $CH_4$, $N_2O$, carbon monoxide, clouds, coarse mode aerosols and other minor gases. 1D-var retrievals such as the AIRS Level 2 products (Susskind et al., 2014) attempt to retrieve all relevant atmospheric and surface variables in order to produce the most accurate temperature and $H_2O$ profiles. The atmospheric $CO_2$ concentration is especially important for AIRS retrievals since most of the radiance measured in the temperature sounding channels is due to $CO_2$ emission. However, it is difficult to separate the $CO_2$ concentration from variations in the temperature profile due to co-linearity of their Jacobians. Consequently, the AIRS Level 2 retrievals instead vary $CO_2$ in the forward model to account for $CO_2$ growth during the mission (Blaisdell, 2019).

The largest radiance trends seen by AIRS are due to the growth rate of $CO_2$ in the atmosphere. Assuming a nominal growth rate of 2 ppm/year, and max sensitivity of AIRS channels of $CO_2$ of 0.03K/ppm, the brightness temperature (BT) shift in AIRS over 16-years is ~1K, or 0.06K/year. Concentrations of atmospheric carbon dioxide have been measured worldwide for many years with extremely high accuracy (Masarie and Tans, 1995; Tans and Keeling) by NOAA Earth System Research Laboratory (ESRL). Averaged yearly, $CO_2$ concentrations are highly uniform globally, with little latitudinal variation in growth rates. Similarly NOAA ESRL also provides a wide network of measurements of $N_2O$ and $CH_4$, which are also relatively uniformly mixed over yearly time-periods. Here we use the high accuracy of the trends in these in-situ measurements of minor gases to determine the stability of a large number of AIRS channels.

SST trends are also extremely well measured and generally referenced to the in-situ ARGO (Argo, 2019) buoy network but interpolated to a full grid using instruments such as the AVHRR. Two SST products referenced to the buoy network are compared to AIRS trends here: (1) NOAA's Optimum Interpolation SST (version 2) (OISST) (Banzon et al., 2016), and (2) the Operational Sea Surface Temperature and Ice Analysis (OSTIA) (Stark et al., 2007), which has been used in the ERA-I Interim Reanalysis (ERA-I) since 2009 (Dee et al., 2011). Prior to Feb. 2009 ERA-I used the NCEP Real-Time Global SST (RTG) product, a precursor to OISST.

AIRS stability is referenced to trends in these minor gases and SST by performing 1D-var retrievals of clear scene radiance anomalies averaged into 40 equal-area latitude bins and 16-day time periods. Comparisons of the retrieved gas concentrations and SST trends, combined with examination of the retrieval residuals, provides a number of powerful tests of AIRS radiometric stability as well as detailed information on AIRS performance changes due to several minor instrument shutdowns that took place occasionally over the mission.

After summarizing the characteristics of the AIRS instrument, and the data used in this work, the retrieval methodology is reviewed with a short discussion of the retrieved temperature profile time series. We follow with stability estimates derived

from the anomaly spectra retrievals of $CO_2$, $N_2O$, $CH_4$, and SST. Although AIRS is most sensitive to the two best in-situ data sets, $CO_2$ and SST, we also compare to retrievals of $N_2O$ and $CH_4$ since they are also relatively well measured and help test the AIRS performance in spectral regions not covered by $CO_2$ and SST. Finally we examine the time series of the anomaly retrieval residuals (BT observed - fit) time series since, together with the anomaly geophysical retrievals, they provide detailed information on AIRS radiances over time, especially the instrument response to various short shutdowns that occurred during the mission.

## 2  AIRS Instrument and Data

Several details of the AIRS instrument design are relevant to the processing performed here and are needed to understand some of the results. AIRS has 2378 spectral channels divided up into 17 different detector arrays. Appendix A gives the nominal wavenumber boundaries of these arrays. Arrays M-11 and M-12 are linear arrays of single photoconductive HgCdTe detectors. The other AIRS arrays are photovoltaic detectors, and each reported detector output is actually some linear combination of two detectors offset from each other in the vertical (not dispersive) direction. The photovoltaic detectors for each AIRS channel are labeled "A" and "B". The relative contributions of A and B detectors can be changed by command to the spacecraft. The majority of these detectors are wired for equal contributions by the A and B detectors, which we denote as A+B detectors. However, some detectors have always been inoperable, or their performance characteristics changed in orbit, so there are a number of A-only and B-only detectors.

The radiometric and spectral characteristics of the A versus B detectors can be slightly different. During the mission, good A+B detectors can suddenly exhibit greatly increased noise when one or the other of the two detectors fails or degrades. In many circumstances the AIRS Project has changed A+B detectors to be either A-only or B-only in order to recover that particular channel, albeit at slightly lower noise levels than if both detectors were working properly. Fortunately, many of the A-only and B-only detectors are in the window regions where AIRS has tremendous redundancy. Unfortunately, the M-10 array which covers the tropospheric $CO_2$ sounding channels also has a good number of A-only, B-only detectors.

Here we avoid any photovoltaic channel that is not A+B, and any channel with a state change during the mission. Although A-only and B-only channels may perform well, many of these single detector channels exhibit drifts over the mission for colder scenes. This is especially apparent in time series of cold scene observations (deep convective clouds) by comparison to similar time series derived from IASI on METOP-1. In addition, we avoid any channels with detector noise above 0.5K NEDT (for a 250K scene). As discussed below in more detail, we also avoid all short wave AIRS channels, meaning channels past 2000 cm$^{-1}$ for our final trend measurements, since we find that the short wave is drifting slightly.

## 3 Radiance/Brightness Temperature Anomalies

### 3.1 Clear Selection

The new AIRS Level-1c radiance product (Aumann et al., 2020) is used in this work rather than the standard L1b product. The Level-1c product provides single-footprint radiance estimates for channels in Level-1b that are not functional, or are extremely noisy. Even high quality Level-1b channels can sometimes "pop" or experience radiation hits that invalidate the measurement. In these extremely rare cases the Level-1c algorithm substitutes an estimated radiance using a principal-components approach. These corrections are rare enough that they have no effect on the long-term trends under study in this work. Level-1c also includes some channels (in-between detector arrays) that do not exist.

More importantly for this work, the radiances in Level-1c have been corrected for small drifts in the channel center frequencies. These drifts are small, but are large enough to have some minor impact on radiance trends. We emphasize that the channels selected for the anomaly retrievals are all valid Level-1b channels, and most have undergone no corrections other than adjusting the radiances back to a fixed frequency scale.

AIRS L1c clear scenes are primarily detected using a uniformity filter. (Throughout this paper the term "scene" refers to a single AIRS nominal 12-by-12 km. footprint or field-of-view.) The BT of each AIRS ocean scene is subtracted from the BT of each of its 8 neighbors for two window channels at 819.3 and 961.1 cm$^{-1}$. A scene is initially deemed clear only if the absolute value of all of these differences, averaged over the two channels, is less than 0.4K. The selected scenes are matched to ERA-I model fields and a simulated clear BT for the 961.1 cm$^{-1}$ channel is computed using a stand-alone version of the AIRS radiative transfer algorithm (Strow et al., 2003) called SARTA, implemented using HITRAN 2008 line parameters. If the difference between the observed and computed clear scene BT values is more than ± 4K the scene is discarded from the clear list. This test mostly removes colder scenes made up of very uniform marine boundary layer stratus clouds. The clear yield and mean zonal radiances are quite insensitive to the exact value of this threshold. The uniformity test is not performed on the first and last of the 135 along-track scans in each AIRS granule since they do not have 8 neighbors and we wanted to avoid cross-granule processing. The total number of clear scenes is limited to ~40,000 daily clear scenes by randomly sub-setting the detected clear scenes, however this daily limit is almost never reached. In this work we only use descending node observations in order to avoid solar and nonLTE contributions to the AIRS radiances in the short wave. After subsetting for descending (ocean) only the total number of clear scenes detected is ~10,000 per day.

The 4K (observed minus computed) BT test removes ~20% of the scenes detected with the uniformity filter. A map of these deleted scenes very clearly shows that they are almost all located along the west coasts of the Americas and Africa, where marine boundary layer stratus clouds commonly occur. The (observed minus computed) BT values for the 1231 cm$^{-1}$ window channel have a nearly Gaussian distribution with a width of ~0.6K. Note that this distribution of biases is well separated from the 4K cutoff used to remove marine boundary layer stratus clouds.

Another important characteristic of this clear subset is the stability of the observing times. If the mean observing time changes during this 16-year time period, trends in the SST could be confused with the diurnal cycle of the SST. Due to the high stability of the AQUA orbit, this is not an issue. The short term day-to-day variations in the mean clear subset times can vary

by several hours. In addition, there is a seasonal variation of several hours in the clear subset. But, these variation are extremely stable, and the total linear drift of the clear subset over the 16-year observing period, for any given latitude bin in the tropics, is ~20 ± 40 (2-$\sigma$) seconds per year, effectively zero.

All observing parameters, on a footprint basis, are saved, such as satellite viewing zenith angle and noise (converted to BT units). In addition, the ERA-I model parameters (temperature, $H_2O$, and $O_3$ profiles, and surface temperature with a spatial resolution of approximately 80 km on 60 levels in the vertical from the surface up to 0.1 hPa) are matched to each clear scene and saved along with their associated simulated L1c radiances. This allows our processing to use simulated rather than observed radiances for testing. The ERA-I profiles are also used to compute the anomaly Jacobians used in the retrievals, and are discussed in detail in Sects. 4.3 and 5.

## 3.2  Clear Scene Characteristics

Figure 1 illustrates the density and location of the clear ocean dataset, averaged over 2012. Retrievals are only performed on

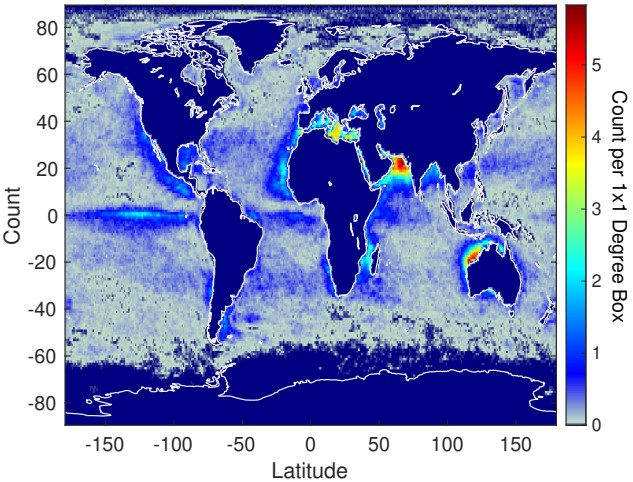

**Figure 1.** Density of AIRS clear ocean scenes for calendar years 2012.

zonally averaged data, which translates into ~44/25 observations/day at -50°/+50°latitude respectively, with a maximum of 200 observations per day at -0.5° latitude. The non-uniform nature of this sampling should be kept in mind when examining temperature, $H_2O$, or $O_3$ trends in that this data set is not necessarily representative of global/zonal climate trends. However, we do assume that the minor gas anomaly trends we are retrieving are uniformly mixed over multi-year time scales. Our anomaly retrieval results show uniform mixing is generally quite accurate over even 16-day time scales.

Figure 2 illustrates the accuracy of ERA-I for this dataset by plotting the (observed - ERA-I) BT bias for 28.4°N. The ERA-I simulated BT used our SARTA RTA, which has a default value of 385 ppm for $CO_2$. This $CO_2$ value is matched in the observations by comparing to AIRS observations for the time period centered around June 2008 when the nominal global $CO_2$ amount was 385 ppm. The window regions (800-1000 cm$^{-1}$) exhibit a bias of ~-0.5K, which is quite small and likely some

combination of instrument bias, evaporative cooling of the ocean surface relative to the ERA-I SST, incorrect ERA-I water vapor column affecting the $H_2O$ continuum, and some cloud-contamination. Sampling errors may contribute to the larger biases in the water region beyond 1300 cm$^{-1}$. A zoom of the bias in the bottom panel of Fig. 2 highlights the low bias in the 700-750 cm$^{-1}$ region which is sensitive to tropospheric $CO_2$, with a mean of ~0.2-0.3K.

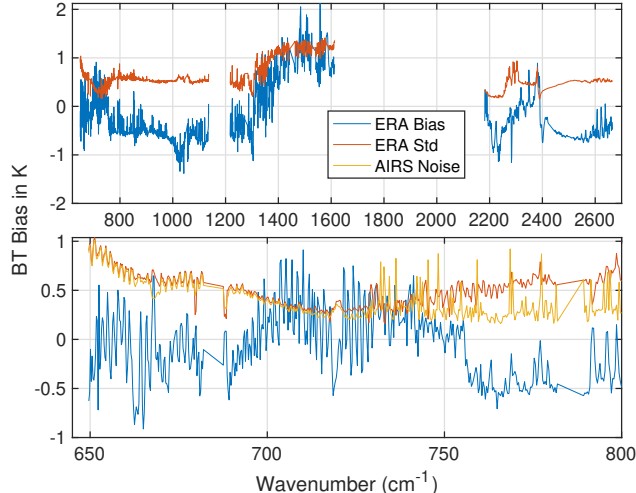

**Figure 2.** Top Panel: (AIRS - ERA-I simulated) BT bias for 28.4$^o$N for a time period centered around June 2008 when the global $CO_2$ amount was ~385 ppm. Bottom Panel: Zoom of top panel. Shows that the region near 700-760 cm$^{-1}$, which is most sensitive to $CO_2$, has a mean bias of ~0.2-0.3K, and a single-footprint standard deviation of ~0.3K. Also shown is the AIRS NEDT, which is barely smaller than the bias standard deviation.

The single-footprint standard deviation of the bias is also shown in Fig. 2, bottom panel along with the average AIRS NEDT for these footprints. The ERA-I bias standard deviation is barely larger than the AIRS noise in this spectral region, indicating that ERA-I temperatures in the mid-troposphere track the AIRS observations very closely with a standard deviation considerably smaller than the AIRS noise. This makes a strong case for the accuracy of the BT Jacobians computed from ERA-I temperature fields.

Figure 3 shows the linear trend for the clear dataset averaged over ±50$^o$ latitude. These BT trends prominently exhibit the growth in $CO_2$ in the tropospheric channels from 700 to 750 cm$^{-1}$, which results in a negative change in the observed BT since increasing $CO_2$ shifts the emission to higher and therefore colder regions of the atmosphere. The growth in $CO_2$ in the stratospheric channels (a positive BT change) below 700 cm$^{-1}$ is roughly cancelled by cooling in the stratosphere. All window channels exhibit warming, with larger values in the shortwave past 2450 cm$^{-1}$. Spectral regions in Fig. 3 that exhibit trends smaller than the 2-$\sigma$ uncertainty are often channels where the BT trends that are due to increasing $CO_2$, $CH_4$, and $N_2O$ are counter-balanced by changes of the opposite sign due to trends in either the atmospheric or surface temperature. These counter-balanced trends are all properly accounted for in the anomaly retrievals given the good agreement between the observed and in-situ minor gas trends. The non-uniform spatial sampling of these clear scenes precludes any general statements about climate

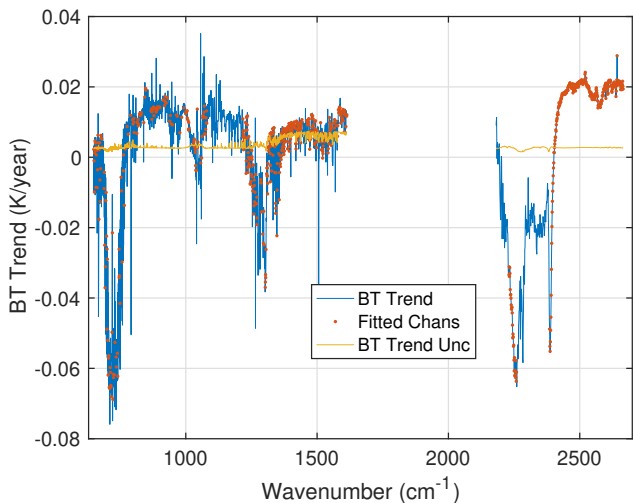

**Figure 3.** Mean BT trends ($a_1$ in Eq 1) averaged over $\pm50^{\circ}$ in $\Delta$ BT/year units. The 2-$\sigma$ uncertainty shown has been corrected for serial correlations in the BT time series. Channels used in the anomaly retrievals are denoted in red, and the BT trend uncertainty is in yellow.

warming, although for these observations we clearly see surface warming in the 800-1250 cm$^{-1}$ region, if the AIRS radiometry is stable. In addition, the effects of much stronger water vapor absorption in the long wave compared to the short wave windows makes definitive inter-comparisons of the BT trends complicated, which is addressed below by doing retrievals on these data.

### 3.3 Construction of Anomalies

The clear scene radiance subset is sorted into 40 equivalent area latitude bins that cover the full -90$^{\circ}$ to 90$^{\circ}$ latitude range and are averaged over every 16 days. This results in a data set for the first 16-years of AIRS that has a size of (40 x 2645 x 365) latitude bins, AIRS L1c channels, and the total number of 16-day averages. The following time-series function was fit to these averaged radiances, $r_{\mathrm{obs}}(t)$, for each latitude and AIRS L1c channel,

$$r_{\mathrm{fit}}(t) = r_o + a_1 t + \sum_{i=1}^{4} c_i \sin(2\pi n t + \phi_i) \tag{1}$$

where $t$ is AIRS mission times in years, $r_o$ is a constant, $c_i$ are the amplitudes of the season cycle and three harmonics, and the $\phi_i$ are their associated phases. At 28$^{\circ}$N, for example, the annual amplitude relative to the mean radiance, $c_1/r_o$, has a median value (taken over channel) of 4.2%. The median amplitudes of the three harmonics terms, $c_2, c_3, c_4$ relative to $r_o$ is 0.32%, 0.45% and 0.23% respectively, all with 2-$\sigma$ uncertainties of ~0.05%. The linear trends $a_1$ are included in the anomaly time-series fits for simple diagnostic purposes, and are not used directly in the anomaly retrievals.

The radiance anomalies, $r_a(t)$ are formed by removing the constant $r_o$, and the sinusoidal terms in Eq. 1, from the observed radiance time series $r_{\mathrm{obs}}(t)$. This can be expressed as

$$r_a(t) = r_{\mathrm{obs}} - \left( r_o + \sum_{i=1}^{4} c_i \sin(2\pi n t + \phi_i) \right) \tag{2}$$

The radiance anomalies $r_a(t)$ were converted to brightness temperature units using

$$y(\nu) \equiv BT_a(\nu, t) = \frac{r_a(\nu, t)}{\dfrac{\partial r(\nu)}{\partial BT(\nu)}}. \tag{3}$$

The 40 x 2645 x 365 array of $BT_a$ vectors are the retrieval inputs $\boldsymbol{y}$ in the retrieval formulation discussed in Sect. 4.

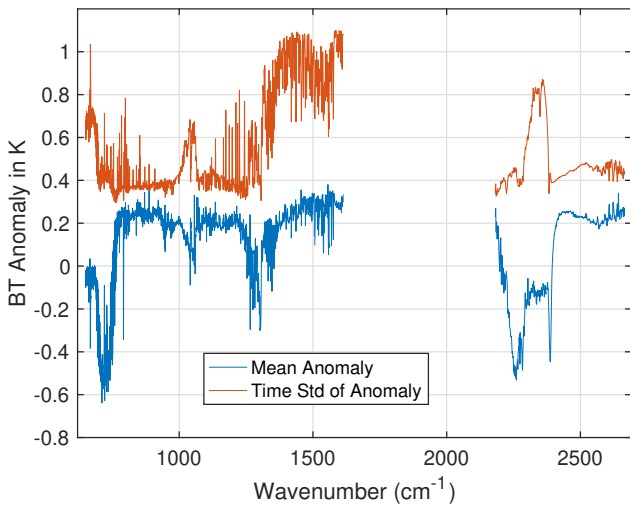

**Figure 4.** Mean and standard deviation of the AIRS BT anomalies for the zonal bin centered at 28.3°N.

The anomaly BT time series mean BT spectra and their standard deviations are shown in Figure 4 for the 28.4°N latitude bin. The BT anomaly is set to zero at the mission start, therefore the mean BT in the channels sensitive to tropospheric $CO_2$ between 700 and 750 cm$^{-1}$ is -0.5K, which then increases by ~1K during the mission. The standard deviation indicates that the SST (and $H_2O$ continuum) vary by ~0.4K during this time period (window region channels from 800-1000 cm$^{-1}$). Some of

this is likely due to changes in sampling from day to day and ERA-I errors in SST and column $H_2O$. Upper-tropospheric water vapor, which dominates the spectral region between 1350-1615 cm$^{-1}$, has the highest variability, which is expected due to both the high temporal variability of water vapor, and our non-uniform sampling.

An example radiance BT anomaly for the 710.141 cm$^{-1}$ channel is shown in Fig. 5, for the same latitude bin. This channel is heavily influenced by the $CO_2$ growth, so the AIRS observed trends are becoming more negative, although there is considerable

noise, again due to weather and sampling. For comparison we also plot the ERA-I simulated BT anomaly, which does not contain the $CO_2$ growth, since it is set to a fixed value of 385 ppm in the simulations. The difference between these two BT anomalies will primarily be due to $CO_2$ growth, and is shown in black. Note that since the ERA-I tracks the atmospheric state quite accurately and most of the time-series "noise" is removed. This helps lend credence to our use of the ERA-I model fields for Jacobian evaluation.

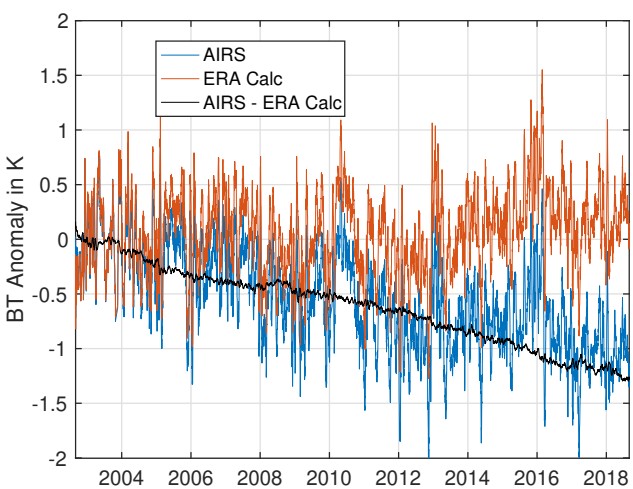

**Figure 5.** Sample AIRS observed and ERA-I simulated BT anomalies for the zonal bin centered at 28.3°N for the AIRS channel centered at 710.14 cm$^{-1}$. The differences in the AIRS and ERA-I anomalies are plotted in black. Note, this difference anomaly is not used in the anomaly retrievals.

## 4 Retrieval Methodology

### 4.1 Approach

Geophysical retrievals are derived from the BT spectral anomalies $\boldsymbol{y}(\nu)$, defined in Eq. 3. Using standard retrieval notation the atmospheric state $x$ is derived from the observations $y$, by minimizing the cost function $J$

$$\mathbf{J} = (\boldsymbol{y} - F(\boldsymbol{x}))^T \mathbf{S}_\epsilon^{-1} (\boldsymbol{y} - F(\boldsymbol{x})) + (\boldsymbol{x} - \boldsymbol{x_a})^T \mathbf{R} (\boldsymbol{x} - \boldsymbol{x_a}) \tag{4}$$

where $\mathbf{S}_\epsilon$ is a diagonal observation error covariance matrix containing the square of the BT noise, $\mathbf{K}$ are the anomaly Jacobians, and $\mathbf{R}$ is a regularization matrix. The retrieved atmospheric state $x$ (the geophysical anomalies) are given by

$$\boldsymbol{x} = \boldsymbol{x_a} + (\mathbf{K}^T \mathbf{S}_\epsilon^{-1} \mathbf{K} + \mathbf{R})^{-1} (\mathbf{K}^T \mathbf{S}_\epsilon^{-1} (\boldsymbol{y} - F(\boldsymbol{x_n}))), \text{where } \mathbf{R} = \mathbf{S}_a^{-1} + \alpha \mathbf{L^T L}, \tag{5}$$

$\mathbf{S}_a$ is the a-priori covariance matrix, and $\alpha \mathbf{L}$ is an empirical regularization constraint using Tikhonov L1-type derivative smoothing. This retrieval approach is standard Optimal Estimation (OE) (Rodgers, 1976) enhanced to include both covariance and empirical Tikhonov regularization in $\mathbf{R}$ (Steck, 2002). Forward model uncertainty is not included in the measurement error covariance. The mathematical approach is very similar to the author's single-footprint AIRS retrieval algorithm (DeSouza-Machado et al., 2018).

A-priori estimates for $\boldsymbol{x_a}(t) \equiv 0$ for T(z), O$_3$(z), H$_2$O(z) and T$_{SST}$ were set to zero. Two approaches were used for the minor gas a-priori estimates. The first approach set $x_a(t) = x_a(t-1)$ where $x_a(t=0) = 0$ for the minor gases, iteratively increasing the a-priori gas amount in time based on the previous 16-day retrieval.

Another approach used the known growth rates in the minor gases (from ESRL) by setting $x_a(t) = g \times (t - t_o)$ for the a-priori minor gas amount, where $g$ is the nominal yearly growth rate for each gas from the NOAA ESRL atmospheric gas trends. For both approaches we set the a-priori covariance to $g$ times one year, the yearly variation in that gas. Nearly identical results are obtained if we used $g$ times five years. The iterative approach for setting the minor gas a-priori produces noisier retrieval anomalies. However, if our retrievals are averaged over $\pm 50^o$ latitude, both approaches produced identical differences compared to in-situ measurements, including error uncertainties. The figures and trend results shown here use the a-priori ramp from the ESRL data, although the figures for the iterative ramp are only distinguishable from what is shown for single zonal retrievals (such as the Mauna Loa and Cape Grim comparisons).

The temperature, $H_2O$, and $O_3$ profile retrievals use 20 atmospheric layers, selected from the AIRS standard 100-layer pressure grid (Strow et al., 2003) by accumulating five of the standard AIRS layers at a time. The lowest layer is about 1.5 km thick, with increasingly wider layers as you go higher in the atmosphere. This layering scheme allows more layers than degrees-of-freedom (DOFs) although it does limit retrievals in the upper-stratosphere. We wish to minimize our sensitivity to the upper-stratosphere since our comparisons to in-situ measurements are made in the troposphere. Consequently we removed all channels peaking above 10 hPa.

Most of the regularization in the retrieval comes from the Tikhonov terms, since we do not want to invoke climatology too strongly for a climate level measurement. Appendix B discusses the profile retrievals, and simulations of these retrievals, in more detail. In summary, after experimentation with Tikhonov regularization we added some a-priori covariance uncertainties in temperature and water vapor of 2.5K and 60% respectively. These are extremely large values for a-priori uncertainties compared to the anomaly variations. For example, the retrieved 400 HPa temperature anomalies shown in Fig. B3 are all less then $\pm 3K$, indicating that the temperature a-priori covariance uncertainty is providing very minimal regularization. This means that almost all of the retrieved temperature variability is coming from the data, and is not damped by the a-priori estimates, a desirable situation for the measurement of climate trends. These a-priori covariance uncertainty terms did, however, improve the profile trends generated in simulation by a slight amount, 3-10%, and thus were retained in our retrieval.

The observation error covariances (noise) are the mean AIRS NEDT for each channel, averaged over 16-days, and then divided by the square root of N, the number of scenes averaged. Originally a fixed value of 0.01K observation noise was used, but we found that this noise value depressed the $CO_2$ anomaly retrievals as they grew in size over time. This problem disappeared once we switched to the true measurement noise values, which are in the range of noise equivalent brightness temperature (NEDT) equal to 0.004K for long wave $CO_2$ channels from 700-750 cm$^{-1}$, about 0.001K in window regions between 800-1250 cm$^{-1}$, and 0.001 to 0.002K in the water band that covers the 1300-1615 cm$^{-1}$ spectra region. These are extremely low noise values, which help explain why the anomaly retrievals have a relatively high number of degrees-of-freedom.

As stated earlier, the profile Jacobians used the ERA-I profiles, which were converted to anomaly profiles for each pressure layer. The minor gas Jacobians were computed using our pseudo line-by-line kCARTA radiative transfer algorithm (Strow et al., 1998; DeSouza-Machado et al., 2019). kCARTA allows for extremely accurate Jacobian calculations, including analytic trace gas and temperature Jacobians. Initial retrievals used a fixed value for the minor-gas Jacobians. However, given the large

increase in the minor gases (10% for $CO_2$), we determined that the minor-gas Jacobians need to be updated as the gas amounts increase. Therefore we used finite-difference Jacobians, computed using the minor gas amount retrieved from the previous time-step during the anomaly retrievals (or from the gas amount estimated using NOAA ESRL in-situ gas amount data). The minor gas profiles used in the Jacobian calculations are from (Anderson et al., 1986). The $CO_2$ profile is essentially constant in ppm until you reach the highest atmospheric layer.

There exists a weak dependence of these retrievals on the ERA-I model fields since we use the ERA-I model fields for the temperature, $H_2O$, and $O_3$ profiles in the profile Jacobians, $\mathbf{K}$. While we could retrieve the atmospheric profiles from the full radiance at each time step and latitude zone, ERA-I is so accurate we do not believe this is needed. Section 5.4 discusses potential errors introduced by using ERA-I for Jacobian evaluation, where they are shown to be extremely small and unimportant.

The direct retrieval of anomalies from the BT anomaly spectra represents a very different approach than normally used in infrared remote sounding. Although the mathematical approach is the same as in single-footprint retrievals (DeSouza-Machado et al., 2018), the often troublesome problem of static measurement and RTA bias errors is largely removed here since instrument calibration and/or absolute RTA biases do not appear in the retrieval process.

## 4.2 Channel Selection

As discussed in Section 2, only channels that remain A+B throughout the mission are used, noting that the designation A+B does not apply to detectors in the M-11 and M-12 long wave detector arrays. Initial retrievals showed that the AIRS short wave detectors are drifting slightly, so these channels are also excluded from the anomaly fits (except for demonstration tests as discussed below). Unfortunately, the use of only A+B detectors greatly restricts the number of available channels in the important long wave $CO_2$ temperature sounding region from 710-780 cm$^{-1}$, where many channels are either A-only or B-only. It is important to weight these channels relatively strongly in the retrieval minimization. Since we also wish to de-emphasize stratospheric contributions to the minor-gas rates only every 5th channel from 650-720 cm$^{-1}$ was included in the retrieval. In addition, any channels in this range with Jacobians that peaked above 10 hPa were excluded.

All channels in the M-5 array were excluded since they have relatively poor radiometric stability (as will be shown later). Several window channels that are sensitive to CFC11 were excluded, although many channels sensitive to CFC12 were included, and CFC12 trends were retrieved. Many $H_2O$ channels were included, since they are mostly A+B and have been stable throughout the mission. After some experimentation, four channels sensitive to $N_2O$ were also excluded since they appear to be behaving significantly out-of-family. Three of these channels are located near the end of the M-4c array, which also exhibits some anomalous frequency shifting behavior (Aumann et al., 2020).

A total of 470 channels remained after this pruning process. These channels are nicely distributed throughout the AIRS spectrum and are easily sufficient for 1D-var retrievals. The nominal number of DOFs for tropical scenes for this channel set are ~6 ozone DOFs, ~8 temperature DOFs, and 12 $H_2O$ DOFs. The larger number of $H_2O$ DOFs is likely due to the large number of $H_2O$ channels used (321 out of 470 channels).

The overall sensitivity of the anomaly retrievals to $CO_2$ is shown is shown in Figure 6 where the mean $CO_2$ Jacobian, averaged over all channels, is plotted. The $CO_2$ sensitivity peaks around 400 hPa, and drops to near zero at the surface. There is some dependence on stratospheric $CO_2$, but stratospheric $CO_2$ trends, especially in the lower stratosphere, should track the tropospheric trends, albeit with growth rates that are slightly influenced by previous years due to age-of-air. This figure also shows the mean $CO_2$ Jacobian if all channels below 700 cm$^{-1}$ are removed (all sensitive to the stratosphere). Retrieval tests using these restrictions are discussed later.

## 4.3 Construction of Jacobians

The relatively high accuracy of ERA-I temperature fields was highlighted previously in Fig. 5 which plots the time dependence of the bias between the observed and simulated BT for this channel. This bias, in black, has very little variability (other than the smooth decrease due to increasing $CO_2$) compared to either the observed or simulated BT values due to the high accuracy of the ERA-I temperature profiles. This is not unexpected in a reanalysis product that assimilates a wide range of in-situ measurements (radiosondes) and satellite measurements (microwave and infrared sounders, including AIRS). In principal we could use the AIRS Level-2 atmospheric state for generating the Jacobians for the anomaly retrievals. However, for the large-scale averaging used in this work errors introduced by the relatively large ERA-I spatial grid compared to AIRS are minimized.

Moreover ERA-I is constrained by a large number of instruments and in-situ measurements for the temperature profile. Monthly mean ERA-I observation minus analysis differences for radiosonde temperatures are below 0.2K throughout the troposphere, rising to 0.3K in the lower stratosphere (Simmons et al., 2014). We note that the statistical accuracy of the AIRS Level-2 algorithm is mainly verified by inter-comparisons with ECMWF forecast/analysis fields (Susskind et al., 2014), which are likely even more stable in a reanalysis product. The AIRS Level-2 retrieved temperature and $H_2O$ global biases relative to ECMWF are very small, well below 0.5K for temperature and 5% for water vapor.

In principle we could have performed 1D-var retrievals on each 16-day averaged BT spectrum in each latitude zone, but given the relatively small biases between ERA-I and AIRS shown in Fig. 2 retrievals will produce minimal improvements to the ERA-I fields. Note that the ERA-I bias in the 700-750 cm$^{-1}$ region with the most sensitivity to tropospheric $CO_2$ is only in the 0-0.5K range. Moreover, 1D-var retrievals using AIRS will also be limited by uncertainties in the AIRS radiometric calibration, which is estimated to be in the 0.2K range (Pagano and Broberg, 2016).

More importantly, since we are only retrieving anomalies, highly accurate Jacobians are unnecessary since the BT variations in the anomalies are so small, especially when applied to trends. A quantitative assessment of errors in our measured anomaly trends from using using ERA-I for Jacobian evaluations is presented in Sections 5.4 and 5.7.

## 4.4 Temperature and Minor Gas Jacobian Co-linearity

A non-standard "correction" is made to the minor gas retrievals that attempts to correct for the co-linearity of the temperature and minor gas Jacobians. We demonstrate that this new approach clearly removes un-physical variability in the $CO_2$ anomaly retrievals. Co-linearity of the temperature and minor gas Jacobians makes it difficult for the retrieval to separate temperature profile variations from variations in $CO_2$, $CH_4$, and $N_2O$. Usually this is managed by constraining the retrievals with accurate

a priori estimates that have small enough covariances to allow some separation of T(z) and $CO_2$ variability. Kulawik et.al. Kulawik et al. (2010) discuss this problem in the context of $CO_2$ retrievals using the NASA EOS-AQUA TES instrument, where they describe the selection of constraints as a way to "determine the partitioning of shared degrees of freedom between
$CO_2$ and temperature".

     Here we take a different approach based on the fact that we have highly accurate simulated anomalies computed from the ERA-I model fields. The simulated anomalies are derived using our SARTA radiative transfer algorithm (RTA) and were generated using constant values for the minor gases throughout the 16-year time period. Except for the minor gas signatures, the ERA-I spectral anomalies are very similar to the observed anomalies since both AIRS calibration errors and RTA errors are
largely removed when forming the anomalies. Fig. 5 shows the excellent agreement between the observed and simulated BT anomalies for the 710.14 cm$^{-1}$ channel. The only major difference in these anomalies is the downward drift in the observations primarily due to the growth of $CO_2$. Note that almost all the high frequency variability in the observed and simulated anomalies is removed when taking their difference, shown in black in Fig. 5, indicating that the ERA-I temperature fields match the AIRS observations very closely.
Given that the ERA-I spectral anomalies are very similar to observed anomalies we can largely determine the effect of the Jacobian co-linearities on the observed $CO_2$ anomaly retrievals by retrieving a (fictitious, or non-existing) $CO_2$ anomaly from the ERA-I simulated BT anomalies using an identical retrieval algorithm. Since the simulated anomalies have a constant value for each minor gas the variations in the retrieved $CO_2$ (or other minor gases) is a measure of the inability of the retrieval to separate the minor gas anomalies from the temperature profile.

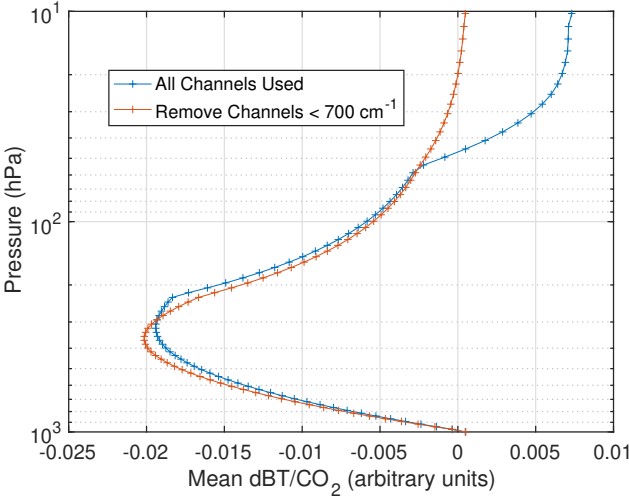

**Figure 6.** Mean of $CO_2$ Jacobians for all channels used in the anomaly retrievals, and the same if all channels below 700 cm$^{-1}$ (stratospheric channels) are excluded.

Figure 7 illustrates this process for; (1) a single latitude bin near -55°lat (with a width of ~4°latitude) in the left hand panel and (2) the average of 30 latitude bins covering ± 50°latitude in the right hand panel. The yellow curve (labelled Simulated) is

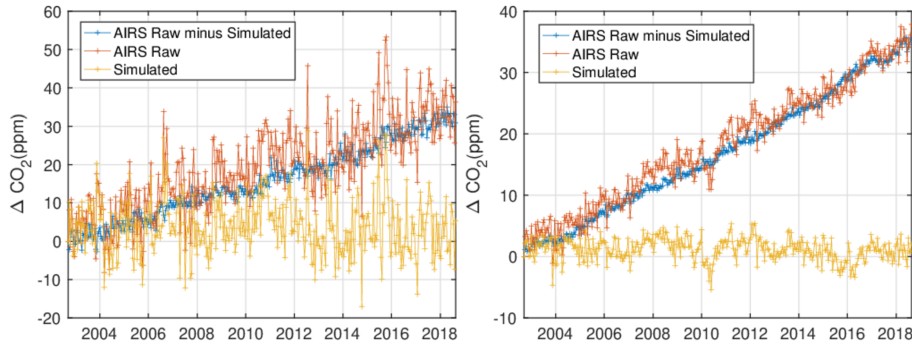

**Figure 7.** Illustration of "noise" removal in the $CO_2$ anomaly retrievals by subtracting the $CO_2$ retrieved from ERA-I simulations from the observed $CO_2$ retrievals. Left: -55° latitude $CO_2$ retrieval. Right: ± 50° latitude average $CO_2$ retrieval.

the retrieved $CO_2$ anomaly derived from the ERA-I simulated anomalies. Although close to zero in the mean, this retrieved $CO_2$ anomaly varies considerably by up to ± 15 ppm. The red curve (labelled AIRS Raw) shows the $CO_2$ anomaly retrieved from the AIRS observations, which has similar variability superimposed on a linear ramp of ~2 ppm/year. The adjusted observed $CO_2$
anomaly is generated by subtracting the simulated from the observed $CO_2$. This is shown in blue (labelled AIRS Adjusted), showing that most of the "noise" has been removed resulting in a very smooth $CO_2$ anomaly curve.

The right hand panel of Fig. 7 shows similar results but using the average of all latitude bins between ± 50°latitude. The co-linearity of the temperature and $CO_2$ Jacobians apparently changes randomly enough with latitude that the simulated $CO_2$ has far less variability than in the left panel. The utility of this approach is nicely illustrated by examining the dip of about 7
340     ppm in the "Simulated" $CO_2$ retrieval in early 2010 for the ± 50°latitude bin. This dip is also visible in the observed anomaly curve (AIRS Raw). The "Simulated" $CO_2$ anomaly is subtracted from the "AIRS Raw" curve to obtain the final observed $CO_2$ anomaly (AIRS adjusted) and it is quite evident that the dip in early 2010 has cancelled out, as desired.

The above adjustments to the $CO_2$ anomaly retrievals have little effect on estimates of AIRS stability over 16 years, as outlined later in Sect. 5.4, although it does increase the statistical uncertainty in the AIRS BT trends by a factor 2.4. More
importantly, the application of these adjustments greatly reduces the apparent noise in the derived $CO_2$ trends, making the detection of instrument shifts in the AIRS BT time series much more sensitive.

## 5     Anomaly Retrievals

### 5.1     AIRS Events

Evaluation of the anomaly retrievals requires some knowledge of the AIRS mission events. Table 1 summarizes the major
events during the AIRS mission that had thermal consequences for either the spectrometer or the focal plane arrays. While most of these events were minor, recent measurements of the AIRS frequency shifts (Aumann et al., 2020) highlight that these events are associated with small shifts in the AIRS frequency scale. These shifts are indicative of very small movements of the

detectors relative to the instrument spectrometer axis and could, for example, slightly alter the detector's view of the blackbody and cold scene. Any small non-uniformities in these calibration looks could affect the absolute radiometry. We will refer to these events during discussions of the anomaly retrieval results.

**Table 1.** Summary of AIRS events that had a thermal impact on either the spectrometer, the focal plane, or both.

| Date | Event |
| --- | --- |
| 10/29/03 | AQUA shutdown lasting for several weeks (solar flare) |
| 01/09/10 | Single event upset, focal plane temperature cycling |
| 03/28/14 | Single event upset, small focal plane cooler variation |
| 09/25/16 | Single event upset, one cooler restart |

## 5.2 Truth Anomalies

The retrieved minor gas anomalies are compared to the NOAA Earth System Research Laboratories (ESRL) monthly mean data derived from in-situ measurements (Tans and Keeling) for the Mauna Loa and Cape Grim site, and for the global mean data for $CO_2$, $N_2O$, and $CH_4$. Monthly anomalies for these in-situ datasets were computed using the same methods used to compute the BT anomalies for consistency. We focus mainly on the global $CO_2$ ESRL anomalies since they are derived from a wide geographical range and sites and carefully merged to avoid local sources. The $N_2O$ ESRL anomalies provide information on AIRS channels in the 1250 -1310 cm$^{-1}$ region that are distinct from the main $CO_2$ channels below 780 cm$^{-1}$. (There are also strong $N_2O$ channels in the short wave band of AIRS.) The $CH_4$ anomalies mostly probe AIRS channels from 1230 to 1360 cm$^{-1}$. There is some concern that $CH_4$ anomaly trends may have more spatial variability than $CO_2$ and $N_2O$, however we find good overall agreement with the ESRL global $CH_4$ trends, and $CH_4$ provides some sensitivity to channels that overlap with $N_2O$, but extend a bit further into the water band.

We focus mostly on the use of $CO_2$ for AIRS stability estimations since $CO_2$ is so well measured and has the largest BT signal in the AIRS spectrum (relative to $N_2O$ and $CH_4$). In addition, the $N_2O$ and $CH_4$ spectra overlap strongly in the AIRS BT spectrum, possibly introducing some retrieval uncertainty relative to $CO_2$. Absolute errors in the ESRL $CO_2$ data are estimated to be ~0.2 ppm (https://www.esrl.noaa.gov/gmd/ccl/ccl_uncertainties_co2.html), with yearly growth rate uncertainties of ~0.07 ppm/year (https://www.esrl.noaa.gov/gmd/ccgg/trends/gl_gr.html). Anomaly growth rate errors averaged over 16 years are likely much lower since yearly sampling errors should diminish over time. Moreover, most absolute errors will not be applicable to the $CO_2$ anomaly, which is a relative measurement. Therefore it is difficult to definitively estimate the ESRL anomaly trend uncertainty. If the yearly growth rate uncertainties of 0.07 ppm/year are random, then the average of 16 of these growth rates would be 0.018 ppm/year, which corresponds to a percentage uncertainty of 0.8% in the anomaly trend.

Estimates for $N_2O$ and $CH_4$ anomaly trend uncertainties using the ESRL stated uncertainties in yearly growth rates, and assuming these are random errors each year, are 3.5% and 2.4%. These larger uncertainties, and the smaller total impact of

these two gases on the AIRS BT anomalies, suggest that the best estimates for AIRS stability are likely derived from the $CO_2$ anomalies.

 ## 5.3 Short Wave Trends

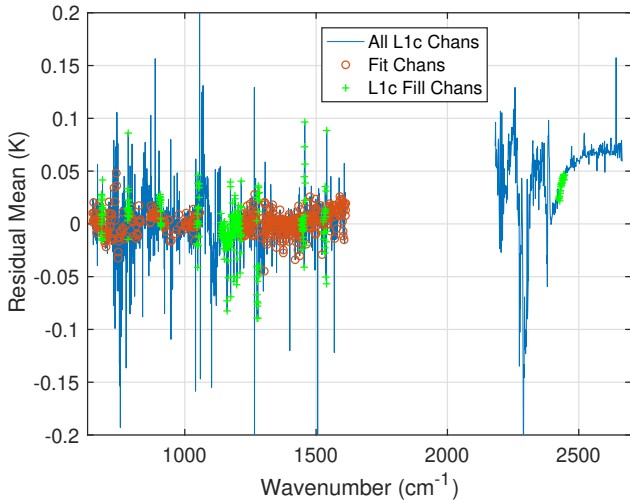

**Figure 8.** Anomaly fit residual, averaged over all 365 16-day time steps for ± 30° latitude. The L1c fill channels have no L1b counterparts and are simulated in the production of L1c. Note the offset in the short wave.

Most of the anomaly retrievals performed here only included AIRS channels located below 1615 cm$^{-1}$, avoiding the short wave channels in the 2181 to 2665 cm$^{-1}$ region. Early retrievals showed that the AIRS short wave channels exhibit a positive trend compared to the longer wave channels. Moreover, anomaly fits to just the short wave channels return SST trends that are significantly larger than both the long wave channels and both the ERA-I (OSTIA) and OISST SST products.

The behavior of the AIRS short wave channel relative to the long wave is easily seen in the anomaly retrieval fit residuals. Figure 8 shows the mean value (taken over the 365 16-day time steps for ± 30° latitude) for the residuals. All AIRS L1c channels are plotted, which includes many bad channels, and channels that do not exist but are filled during L1c creation (Aumann et al., 2020). The channels selected for the anomaly fits (see Sect. 4.2) are shown in red circles. The fit residuals for channels used in these retrievals are almost all well below 0.02K. However, the short wave channels show anomalies inconsistent with the long 390    wave of up to ~0.07K in the window channels past 2450 cm$^{-1}$.

The anomaly retrievals can respond to drifts/offsets in the AIRS radiances by retrieving geophysical variables ($CO_2$, temperature, etc.) that vary incorrectly in time. Alternatively, un-physical changes in the radiances could also be reflected in larger non-zero fit residuals. This could happen when the forward model Jacobians cannot model time-dependent radiance errors, especially for jumps in the radiometric calibration that happen due to AIRS events (shutdowns). One way to examine this 395    possibility is to look for any remaining trends in the anomaly fit residuals. These are shown for the same data set used in Fig. 8

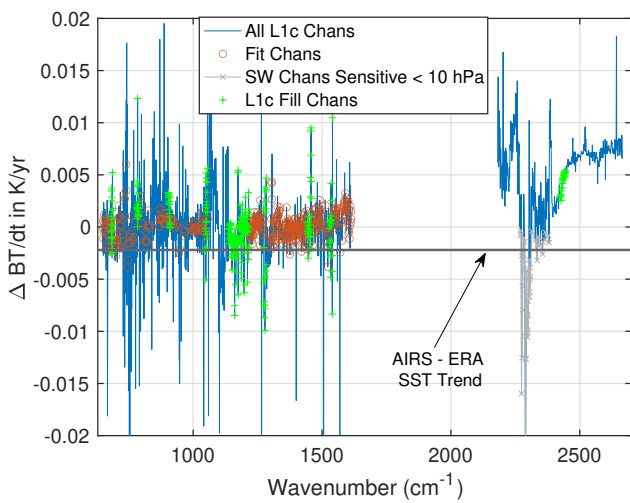

**Figure 9.** Linear trends in the anomaly fit residuals, averaged over all 365 16-day time steps for ± 30° latitude. Note the linear trend in the short wave in these fit residuals. Also shown is the trend difference (ERA-I SST - AIRS SST) for these data.

in Fig. 9. Most of the channels used in the anomaly fits have residual slopes below 0.002K/year, although careful examination of the residual time series for particular channels can exhibit jumps associated with AIRS shutdowns.

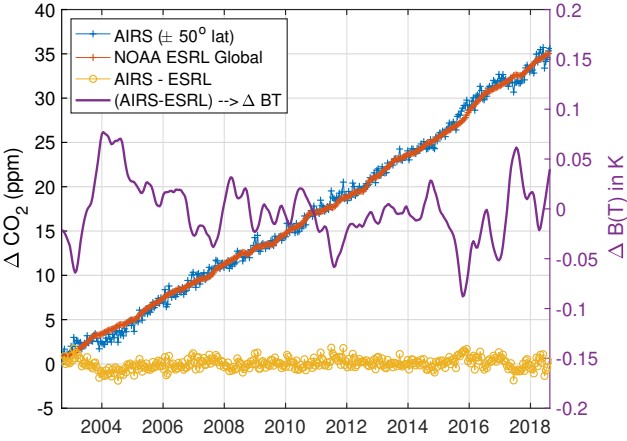

**Figure 10.** Retrieved $CO_2$ anomalies compared to ESRL global in-situ data. The $CO_2$ anomaly difference between AIRS and ESRL is shown in yellow. The magenta curve is that difference converted into BT units.

The main observation in Fig. 9 is a clear positive trend in the short wave relative to the longer wave channels used in the retrievals. The (AIRS - ERA) SST trend plotted as a solid horizontal line in this figure (discussed in Sect. 5.7) shows that the AIRS short wave trends are more different from the ERA-I SST trends than the long wave channels. Most of the short wave

channels, including those in the mid-troposphere, exhibit positive trends relative to the long wave, except for some channels that are peaking very high in the stratosphere, below 10 hPa, that are marked in gray.

Consequently, unless otherwise noted, all the remaining results presented here avoid short wave channels, and use the channel set (470 channels) denoted in these figures.

### 5.4 CO$_2$ Anomaly Retrievals

Figure 10 shows the retrieved CO$_2$ anomalies averaged over ± 50° latitude in blue and the ESRL global anomaly product in red. The correspondence over time is excellent. The AIRS minus ESRL anomaly differences are shown in yellow. In order to

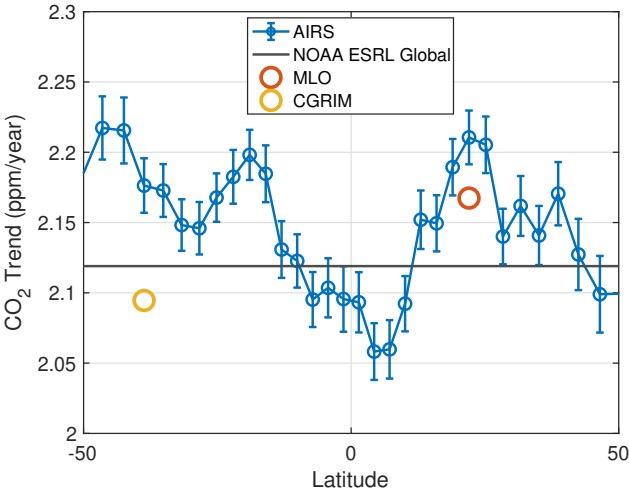

**Figure 11.** Observed linear trend in the AIRS CO$_2$ anomalies versus latitude, compared to NOAA ESRL Mauna Loa (MLO), ESRL Cape Grim (GCRIM), and the ESRL global CO$_2$ product trends (black line).

convert the variation in the gas anomalies to an equivalent AIRS BT anomaly temperature we computed anomaly retrievals with the observed AIRS BT anomaly spectra modified by a 0.01K/year ramp, for all channels. This 0.01K/year ramp is divided by the resulting changes in the CO$_2$ anomaly linear trends (ppm/year) to obtain the sensitivity of the retrieval to a trend in the AIRS radiances, in K/ppm. For CO$_2$ this sensitivity is -0.073K/ppm. This is about 2X larger than the largest column Jacobians in the AIRS spectra, which have a value of ~0.030K/ppm. This is not unexpected, since the CO$_2$ column measurement is partially a relative measurement, especially for weak CO$_2$ channels in the window region where the absolute BT errors are mostly accounted for by (incorrect) adjustments in the SST that minimize the effect of the 0.01K/year applied ramp. It is also possible that the temperature profile could also adjust to minimize sensitivity of the ramp on the CO$_2$ ppm values. In addition, this sensitivity estimate assumes all AIRS channels are drifting, which is clearly an approximation given the results shown here.

The magenta curve in Fig. 10 is the (AIRS minus ESRL) anomaly differences converted to BT units using the -0.073K/ppm sensitivity factor. This curve has been slightly smoothed for clarity. The right-hand side vertical axis shows the variations in

**Table 2.** Slope of the (AIRS - ESRL) $CO_2$ anomalies in ppm/year units.

| Data Set $CO_2$ | Mean Trend Difference (ppm/year) | Uncertainty in Trend (ppm/year) |
|---|---|---|
| Global | 0.032 | 0.012 |
| Mauna Loa | 0.033 | 0.023 |
| Cape Grim | 0.056 | 0.020 |

this curve in BT units. Most of the BT variability is within ± 0.05K, however a transition in BT in late 2003 is larger. This larger transition is likely due to the Nov 2003 shutdown of the AQUA spacecraft. The AIRS channel center frequencies were shifted due to this shutdown (Strow et al., 2006) and were subsequently corrected in the AIRS L1c product (Aumann et al., 2020; Manning et al., 2019). In addition, as reported in (Strow et al., 2006) interference fringes in the AIRS entrance filters shifted after the Nov. 2003 AQUA shutdown because AIRS was restarted at a slightly different spectrometer temperature. The

fringes change the AIRS spectral response functions, which has not yet been corrected in the AIRS L1c product radiances.

Figure 11 illustrates the differences between the AIRS and ESRL $CO_2$ linear growth rates. The growth rates for both our $CO_2$ retrievals and the ESRL $CO_2$ time series were computed by re-using the fitting function in Eq. 1, but now applied to the retrieved $CO_2$ anomalies, ie.

$$CO_2(t) = CO_2(t=0) + b_1 t + \sum_{i=1}^{4} d_i \sin(2\pi n t + \phi_i) \tag{6}$$

where $b_1$ are the $CO_2$ trends in ppm/year. Later this equation will be used to fit the $N_2O$, $CH_4$, and SST anomalies, instead of the $CO_2$ anomalies as shown here.

Figure 11 plots the fitted values for the AIRS growth rates (the $b_1$ term in Eq. 6), computed as a function of latitude. The $CO_2$ growth rates are not completely uniform from year-to-year, so Eq. 6 cannot perfectly fit the trend data. However, it provides a convenient metric for inter-comparing these two $CO_2$ anomalies. Note that the error bars shown for AIRS are slightly over-

estimated because of the fact that Eq. 6 does not perfectly fit the slightly non-linear anomaly curve. The error estimates are 95% confidence intervals and they have been corrected for serial correlations in the anomaly time series using the popular lag-1 auto-correlation approach detailed in (Santer et al., 2000).

The Mauna Loa and Cape Grim growth rates are also shown, also derived using Eq. 6, as is the ESRL global rate, indicated by the dark black horizontal line. If the 16-year in-situ rates indeed have an estimated error of 0.018 ppm/year (assuming

the 0.07 ppm/year uncertainties in the ESRL rates are random), then AIRS is in close agreement with ESRL averaged over latitude. The latitude dependence of the AIRS derived rates appear to have clear latitudinal dependencies, with lower rates near the ITCZ and higher rates in regions of descending air. We do not examine this latitude dependence in this work, not only is it small, it could also be related to small inaccuracies in our retrieval algorithm.

Since the $CO_2$ linear growth rate measurements are not sensitive to year-to-year variability in the $CO_2$ anomaly, we instead

use the (AIRS - ESRL) global anomaly differences shown in Fig. 10 to quantify the AIRS stability. Any linear trend differences between the AIRS and ESRL $CO_2$ in Fig. 10 are quantified by fitting the (AIRS - ESRL) $CO_2$ anomaly differences to Eq. 6.

**Table 3.** Slope of the (AIRS - ESRL) $CO_2$ anomalies in K/Decade units. Trend differences for various modifications of our retrieval algorithm are shown, see the text for details. Note that Baseline is the algorithm configuration detailed in the text and used for inter-comparisons.

| $CO_2$ Test | Mean Trend Difference (K/Decade) | Uncertainty in Trend (K/Decade) |
|---|---|---|
| Global | | |
| Baseline | -0.023 | 0.009 |
| Baseline (no $CO_2$ adjustment) | +0.019 | 0.022 |
| No Strat | -0.034 | 0.008 |
| No Cov Reg. | -0.043 | 0.009 |
| No $\nu$ Cal. | -0.059 | 0.010 |
| Shortwave Only | +0.070 | 0.009 |
| ERA-I T(z) | +0.060 | 0.035 |
| Mauna Loa | | |
| Baseline | -0.024 | 0.017 |
| Cape Grim | | |
| Baseline | -0.040 | 0.020 |

Table 2 summarizes any trend in AIRS relative to ESRL by tabulating the $b_1$ terms from the fit for the ESRL global, Mauna Loa, and Cape Grim sites. The uncertainties are as before, 95% confidence intervals corrected for lag-1 auto-correlations. As one might expect, the global trends agree the best, and Cape Grim the worst. The higher errors for Cape Grim may be related

to our clear subset having fewer samples at -40° latitude relative to the 20° latitude zone occupied by Mauna Loa. These mean differences are extremely small, corresponding, for global, to 1.5 ± 0.6% trend differences.

Table 3 shows the conversion of the $CO_2$ ppm trend differences to equivalent BT differences using the -0.073 K/ppm sensitivity conversion. The baseline entry, first line of the table, represents the final configuration for the anomaly retrievals and is our best estimate for the differences between the ESRL and AIRS $CO_2$ anomaly trends, -0.023 ± 0.009 K/decade. This

is an exceedingly small trend difference. While suggesting that AIRS is extremely stable, for channels sensitive to $CO_2$ and temperature, systematic errors may be larger than the differences reported here. Our estimate of the ESRL global anomaly trend uncertainty discussed in Section 5.2, 0.8%, is equivalent to 0.017 ppm/year. The AIRS minus ESRL global trend difference shown in Table 2 is about 2X times larger than this estimate for the ESRL uncertainty and slightly larger than the statistical uncertainty in this trend difference. In BT units, this potential uncertainty in the ESRL global $CO_2$ anomaly trend is ~0.012

K/Decade.

The sensitivity of these results to uncertainties in the Jacobians are derived from the second partials derivative of BT as follows,

$$M_{unc} = \frac{\partial}{\partial X}\left(\frac{\partial BT}{\partial Y}\right) \times X_{unc} \times Y_{meas} = \left(\frac{\partial^2 BT}{\partial X \partial Y}\right) \times X_{unc} \times Y_{meas} \tag{7}$$

where $M$ is the quantity being measured (here, $CO_2$ anomalies and trends), $X_{unc}$ is the uncertainty in the profile variables used to compute the Jacobians, and $Y_{meas}$ is either the maximum anomaly or the mean trend measured for $Y$. These are quantified in Table 4.

The first entry accounts for errors in $\partial BT/\partial CO_2$ due to uncertainties in the $CO_2$ spectroscopy. The HITRAN database (Gordon et al., 2017) reports uncertainties in the $CO_2$ line strengths of 1-2%. These uncertainties would translate into the same percentage error in the Jacobians. In addition, atmospheric spectra are sensitive to line widths, line shape, line mixing, often at temperatures that are not measured in laboratory spectra. Characterizing the combination of these errors is essentially impossible, so here we assume a 1% uncertainty in the $CO_2$ Jacobians, using the line strength uncertainty only. The maximum $CO_2$ anomaly error occurs at the end of the time series when the $CO_2$ anomaly is highest (35 ppm). Therefore the max anomaly error is 1% × 35 ppm = 0.35 ppm. Using the retrieval sensitivity of -0.073K/ppm, this translates into an effect max error in the BT anomaly error of 0.026K. Dividing this anomaly uncertainty by the 16 year time period under study gives a trend uncertainty due to $CO_2$ spectroscopy errors of 0.016 K/decade as shown in Table 4. This value is slightly larger than the statistical uncertainty in the baseline $CO_2$ trend shown in Table 3, and slightly smaller than the derived trend differences versus ESRL $CO_2$.

The second entry in Table 4 lists estimated uncertainties in the $CO_2$ anomalies and trends (converted to BT units) that could arise due to errors in the ERA-I temperature profile. The second partial derivative was computed with finite differences using a fixed temperature offset for all levels and then summed over all levels, a worst case scenario. The mean of these second order derivatives, taken over the retrieval channels in the ~700-750 cm$^{-1}$ spectral region that has high sensitive to $CO_2$, represents an effective scalar value for $\partial^2 (BT)/(\partial X \partial Y)$ in Eq. 7. This term is multiplied by an assumed uncertainty in the ERA-I temperature of 0.5K and by the maximum anomaly value of 35 ppm to obtain a maximum uncertainty of 0.0035K in the $CO_2$ anomaly. The maximum effect on the $CO_2$ trend is again this value divided by 16 years giving an uncertainty of 2.2 × 10$^{-3}$ K/decade, an insignificant uncertainty. Note that our assumed uncertainty of 0.5K is higher than ERA-I error estimates discussed in Section 4.3.

Clearly the estimated 1% uncertainty in the $CO_2$ spectroscopy is the dominant source of error in our $CO_2$ retrievals. If the ESRL 0.8% uncertainty is combined in quadrature with the 1% HITRAN uncertainty, a total minimum expected uncertainty in the $CO_2$ anomaly trends is 1.3%. This translates to a BT uncertainty of 0.02 K/decade, close to our derived mean trend difference between AIRS and ESRL based on the $CO_2$ anomaly measurements. This may be a more accurate uncertainty estimate for this measurement rather than the 0.009 K/Decade statistical uncertainty derived from fitting the AIRS minus ESRL anomalies.

The second entry in Table 3 lists the mean trend difference and its uncertainty if the adjustment for co-linearity discussed in Sect. 4.4 is not applied, which leads to a larger trend uncertainty by a factor of 2.4. The resulting trend difference is somewhat smaller, but with a different sign. The baseline retrievals with and without the co-linear $CO_2$ adjustments do not quite overlap within their respective $2\sigma$ uncertainties, missing statistical agreement by 0.013 K/Decade, which is relatively small. However, based on the discussion in Sect. 4.4 we believe that the application of the co-linear $CO_2$ adjustment improves the accuracy of the AIRS $CO_2$ anomaly.

**Table 4.** Anomaly and trend error estimates for $CO_2$ and SST due to uncertainties in BT Jacobians via their second derivatives with respect to possible ERA-I uncertainties. As noted, the maximum effect on the $CO_2$ anomalies would be at the time of the largest anomaly, which is at the end of our time series in Aug. 2019. See the text for details.

| Jacobian | Sensitivity | Uncertainty | Max Effect on Anomaly | Effect on Trend |
|---|---|---|---|---|
| $\frac{\partial BT}{\partial CO_2}$ | $\frac{\partial^2 BT}{\partial CO_2^2}$ | 1% $CO_2$ Spectroscopy | 0.026K (in Aug. 2019) | 0.016 K/decade |
| | $\frac{\partial^2 BT}{\partial T_{air} \partial CO_2}$ | 0.5K T profile | 0.0035K (in Aug. 2019) | $2.2 \times 10^{-3}$ K/decade |
| $\frac{\partial BT}{\partial T_{SST}}$ | $\frac{\partial^2 BT}{\partial T_{SST}^2}$ | 0.5K $T_{SST}$ | $4.0 \times 10^{-4}$ | $9.6 \times 10^{-5}$ K/decade |
| | $\frac{\partial^2 BT}{\partial T_{air} \partial T_{SST}}$ | 0.5K T profile | $8 \times 10^{-4}$K | $1.8 \times 10^{-4}$ K/decade |
| | $\frac{\partial^2 BT}{\partial T_{H_2O} \partial T_{SST}}$ | 10% $H_2O$ column | 0.02K | $4.5 \times 10^{-3}$ K/decade |

Table 3 also shows the results of a number of fit testing the sensitivity of the retrievals to various retrieval alternatives. The "No Strat" entry removed all channels that primarily sense the stratosphere by removing all channels below 700 cm$^{-1}$. Fig. 6 shows how this modifies the mean $CO_2$ Jacobian used in the retrieval, essentially removing all sensitivity to $CO_2$ above 60 hPa. Unfortunately channels above 700 cm$^{-1}$ have some residual sensitivity to $CO_2$ in the stratosphere, and removing channels below 700 cm$^{-1}$ may make it more difficult to properly minimize the retrieval residuals for some channels above 700 cm$^{-1}$. If $\mathbf{S}_a$ is completely removed, removing a-priori profile regularization, the $CO_2$ anomaly trend difference increases by a factor of two. Removing the L1c frequency calibration adjustments increases the anomaly trend differences by nearly a factor of three, and changes their sign. If only short wave channels are fit (excluding channels that peak above 10 hPA, and some channels sensitive to both carbon monoxide), the mean trend differences are more than three times larger than the baseline, again with a sign change.

The last test, labeled "ERA-I T(z)", examines the need for performing simultaneous retrievals of temperature profiles while retrieving the $CO_2$ anomalies by using the ERA-I temperature profiles anomalies, instead of fitting for them from the observed anomalies. This test increased the anomaly differences between AIRS and ESRL by almost a factor of three, with a significant increase in the uncertainty of the trend, giving 0.35 K/decade instead of close to 0.009 K/decade for the baseline.

Table 3 also shows the Mauna Lao anomaly difference, which is close to the global result, although accompanied by a higher uncertainty of 0.017 K/decade compared to the 0.009 K/decade for the global anomaly. Cape Grim anomaly differences are almost two times higher than the global trend differences, but this is not surprising given the much lower number of observations at that latitude.

The retrieved AIRS global $CO_2$ anomalies did exhibit a small seasonal pattern for latitudes above 40° N of with an amplitude of ~0.5 ppm. This is due to the residual of the seasonal cycle of $CO_2$ that is not completely removed when constructing the BT anomalies.

Note that radiometric shifts or drifts in the AIRS BT time series could be either reflected in incorrect geophysical trends, or partially buried in the anomaly fit residuals. The high quality of the anomaly retrievals for $CO_2$ and the small fit residuals for

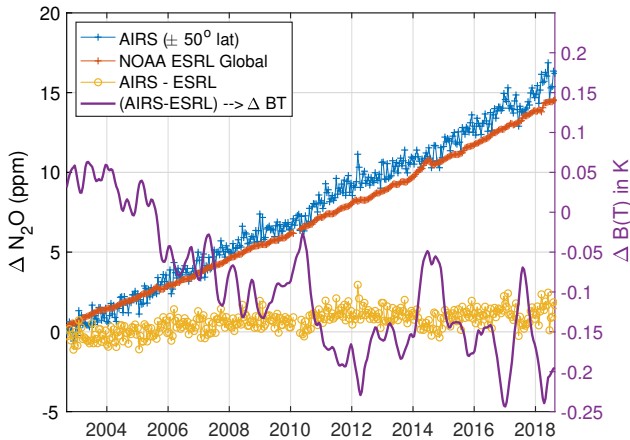

**Figure 12.** Retrieved $N_2O$ anomalies compared to ESRL global in-situ data. The $N_2O$ anomaly difference between AIRS and ESRL is shown in yellow. The magenta curve is that difference converted into BT units.

**Table 5.** Slope of the (AIRS - ESRL) $N_2O$ anomalies in K/Decade units.

| Data Set $N_2O$ | Mean Trend Difference (K/Decade) | Uncertainty in Trend (K/Decade) |
|---|---|---|
| Global | -0.141 | 0.012 |
| Mauna Loa | -0.200 | 0.030 |
| Cape Grim | -0.080 | 0.033 |

$CO_2$ channels strongly suggest that the AIRS blackbody is extremely stable, at least for long and mid wave A+B channels. The SST retrievals discussed later reinforce this conclusion. However, we do see evidence of radiometric shifts due to discrete AIRS events (especially for $N_2O$ and $CH_4$) that might be amenable to correction. Future work will include careful examination of both the anomaly retrievals and their residuals, likely in an iterative fashion, in order to determine what channels are responsible for unphysical shifts in the anomaly products.

### 5.5 $N_2O$ Anomaly Retrievals

The $N_2O$ retrieved anomaly time series is shown in Fig. 12 and primarily senses the 1240-1325 cm$^{-1}$ spectral region. Clearly the observed $N_2O$ anomaly is growing slightly faster than the ESRL values. The $N_2O$ anomalies are converted to equivalent BT variations just as for $CO_2$, but with a derived sensitivity of 0.140 K/ppb. Table 5 tabulates the derived trend for the (AIRS minus ESRL) anomaly by fitting the difference to Eq. 6, and then converting to BT units.

The trend differences here are much larger than for $CO_2$. Examination of either the AIRS minus ESRL anomalies in ppb, or their equivalent in BT units (left hand y-axis) suggest that two unphysical steps might be present in the time series, one in

mid-2005 and another one in mid-to-late 2010. Unfortunately, these steps do not closely coincide with AIRS events, possibly appearing more than one year after the Nov. 2003 event and and slightly less than one year after the Jan. 2010 event.

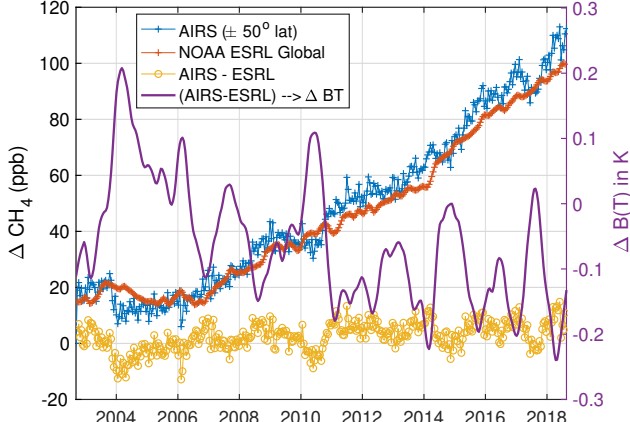

**Figure 13.** Retrieved $CH_4$ anomalies compared to ESRL global in-situ data. The $CH_4$ anomaly difference between AIRS and ESRL is shown in yellow. The magenta curve is that difference converted into BT units.

**Table 6.** Slope of the (AIRS - ESRL) $CH_4$ anomalies in K/Decade units.

| Data Set | Mean Trend Difference | Uncertainty in Trend |
|----------|-----------------------|----------------------|
| $CH_4$ | (K/Decade) | (K/Decade) |
| Global | -0.107 | 0.024 |
| Mauna Loa | -0.062 | 0.039 |
| Cape Grim | -0.100 | 0.037 |

To illustrate the effect of these two discrete shifts on the anomaly trend differences we empirically introduce a step in our retrieved $N_2O$ time series of -0.6 ppb on July 1, 2005 and another step on Jan. 18, 2010 of -0.5 ppb. The trend difference between this empirically modified time series and ESRL, in BT units, becomes -0.022 ± 0.009 K/decade, very similar to the $CO_2$ trend differences. The main point of this exercise is to illustrate that just two small discrete radiometric shifts could be responsible for the higher trend differences between AIRS and ESRL for $N_2O$. More work is needed to map these discrete non-physical events in the retrieved $N_2O$ anomaly time series back into steps in the AIRS BT time series. The hope is that careful examination of the anomaly time series residuals during this process would highlight specific channels (or cluster of channels) that are behaving non-physically.

### 5.6 $CH_4$ Anomaly Retrievals

The $CH_4$ retrieved anomalies have some similarities to the $N_2O$ anomalies, since the spectra of both gases occur in the same general spectral region. The $CH_4$ the region of sensitivity is ~1210-1380 cm$^{-1}$. Figure 13 shows the $CH_4$ results using the same

approach as for $CO_2$ and $N_2O$. The ppb to BT conversion for $CH_4$ was measured to be 0.023 K/ppb, significantly lower than for $CO_2$ or $N_2O$, although total BT trend due to $CH_4$ is only marginally lower than $CO_2$ and $N_2O$.

The high variability of atmospheric $CH_4$ growth is well known, as can be seen in the ESRL curve in Fig. 13. The AIRS derived anomalies follow that variable growth rate quite nicely overall. It should be noted that the ESRL $CH_4$ curve is more variable than $CO_2$ and $N_2O$, and may be less uniform globally, making $CH_4$ a less ideal gas for testing AIRS stability. However, the AIRS minus ESRL anomaly differences are valuable in that they, like $N_2O$, highlight discrete jumps that can often be identified with AIRS events, such as late 2003 (biggest jump), early 2010, and possibly in early 2014. The positive jump in the $CH_4$ anomaly difference near March 2014 also coincides with a jump in the $N_2O$ anomaly difference, both taking place after the March 2014 event. However, this apparent jump seems to fade within one year for both gases. We believe this might be caused by AIRS frequency shifts that occurred in the M-4a and M-4c detector modules after this event. Those frequency shifts appeared to disappear within one year, and at present they are not corrected for in the AIRS L1c product.

Table 6 lists the trend differences between AIRS and ESRL for $CH_4$, showing trends differences that similar to those for $N_2O$, presumably since both gases occur in the same spectral region.

## 5.7   SST Retrievals

The SST anomaly retrievals are compared to the ERA-I supplied SST (mostly OSTIA) and to NOAA's OISST operational SST product. Although both of these SST products are tied to the ARGO floating buoy network, they are gridded SST products using interpolation derived from satellite data such as AVHRR.

A recent study (Fiedler et al., 2019) compared various SST products to the buoy network and found differences for OSTIA of 1.1 mK/year, and 7.8 mK/Year for OISST. This establishes a rough estimate of the differences in these products when evaluating them relative to our retrieved SST anomalies.

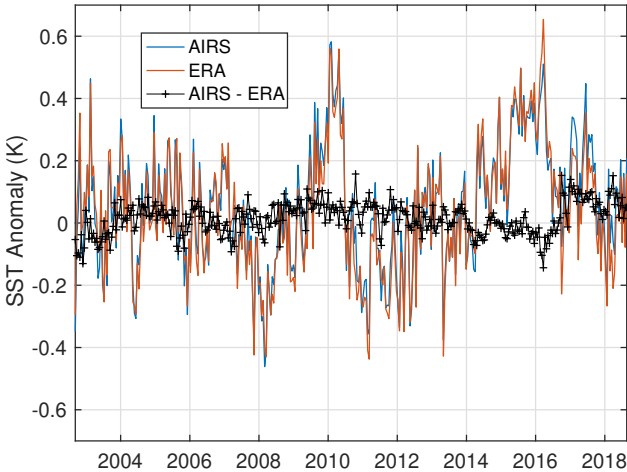

**Figure 14.** Tropical (± 30°) SST anomalies retrieved from AIRS compared to the ERA-I anomalies. The black curve is the difference between the AIRS and ERA-I anomalies.

Figure 14 plots time series of our retrieved SST anomaly and the co-located ERA-I SST (mostly OSTIA) anomaly, averaged over ± 30° latitude, where these products are expected to be most accurate since most buoy's are located in the tropics. The AIRS SST trend derived from this time series is 0.096 ± 0.046 K/decade. The AIRS and ERA-I 16-day averaged anomalies agree very closely, their difference is shown in black. A zoom of the AIRS minus ERA-I SST anomaly is shown in Fig. 15 to highlight their differences. Steps in these differences are possibly evident near the end of 2003 and especially near the end of September 2016 when AIRS had a cooler-restart.

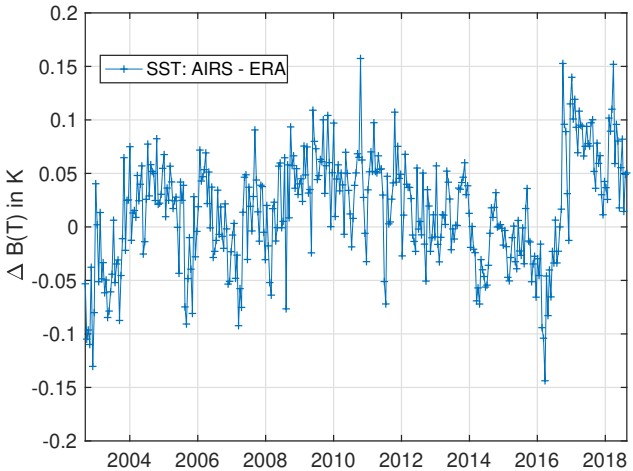

**Figure 15.** Zoom of Fig. 14 that highlights the shift in the AIRS - ERA-I SST anomaly presumably due to the AIRS Sept. 25, 2016 cooler restart. A small shift is also seen at the date of the Nov. 2003 AQUA shutdown.

Table 7 summarizes the AIRS minus (ERA-I and OISST) anomaly trend differences, computed using Eq. 6. The trend differences are quite small for both SST products. The (AIRS minus ERA-I) trend has the same magnitude as the trend derived using $CO_2$, but with the opposite sign. Overlap of the $CO_2$ and ERA-I SST within their stated uncertainty estimates is missed by 0.01K/decade, which is very small. The $CO_2$ and OISST trend estimates miss overlap by slightly more, 0.02K/decade. However, this overlap difference is small compared to the differences between OISST and the buoy network reported by (Fiedler et al., 2019). Overall the excellent agreement of these two extremely independent assessments ($CO_2$ versus SST) to within 0.02K/decade is very encouraging given the complexity of the $CO_2$ measurement and the uncertainties in the SST product trends.

Comparisons between AIRS-derived SST and ERA-I or OISST products will contain biases due to time aliasing between the AIRS observations and daily means used in the SST products. Although these time dependent biases can have random and seasonal variations of several hours the observed linear drift in the AIRS local observing time over the 16-year observation period was less than one minute/year, far too small to introduce any drifts in the AIRS SST relative to the ERA-I or OISST daily averages.

Uncertainties in the SST anomaly retrievals due to our use of ERA-I fields for the evaluation of the SST Jacobians were estimated using the same approach for the $CO_2$ anomaly retrievals. The BT Jacobians (dBT/dSST) for channels sensitive to

SST depend on accurate values for the SST itself, the air temperature profile, and most importantly the $H_2O$ profile, especially in the lower troposphere. We computed the partial derivatives of the BT Jacobians with respect to all three of these variables,
again using finite differences and a constant offset for the air temperature profile and constant percentage offsets for the $H_2O$ profile. The partial derivatives were averaged for all AIRS channels used in our retrievals in the 800-1235 cm$^{-1}$ region that is sensitive to surface temperature. The uncertainties assumed in the ERA model fields ($X_{unc}$ in Eq. 7) are listed in column three of Table 4 and are likely higher than the estimated uncertainties summarized in Section 4.3. The uncertainties in the BT Jacobians are then multiplied by $Y_{meas}$ in Eq. 7 which is either 0.4 K (the maximum SST anomaly, see Fig. 14), or
0.0096K/year (our retrieved trend in SST).

The results shown in columns four and five of Table 4 clearly indicate that using ERA profile fields for estimated BT surface temperature Jacobian is extremely accurate. The highest uncertainties are due to $H_2O$, but even these are far below the statistical uncertainties shown in Table 7.

**Table 7.** Slope of the (AIRS - (ERA/OISST)) SST anomaly differences.

| Data Set | Mean Trend Difference (K/Decade) | Uncertainty in Trend (K/Decade) |
|---|---|---|
| (AIRS - ERA-I) | 0.022 | 0.012 |
| (AIRS - OISST) | 0.034 | 0.021 |

Aumann (Aumann et al., 2019) recently compared the 1231 cm$^{-1}$ AIRS channel trends to RTGSST, a precursor to OISST.
He used a statistical approach to remove trends in water vapor that affect the 1231 cm$^{-1}$ channel radiances, which he concedes could introduce artifacts if there is a shift in the mean vertical distribution of water vapor. Our approach does not contain this limitation in principle, although we have not carefully examined the retrieved water vapor trends, mainly because there is no truth for comparison. An intercomparison of our results to his are not strictly possible since we used different SST products for truth and our SST anomalies used many channels. However, the trend of the 1231 cm$^{-1}$ channel in our retrievals can be
derived by adding the slope of our fit residual for the 1231 cm$^{-1}$ channel (-0.7 mK/year) to our derived SST trends for ERA-I and OISST. Using Aumann's units of mK/year, the result is a trend of 1.5 mK/year and 2.7mK/year for ERA-I and OISST respectively, with respective uncertainties of 1.2 and 2.1 mK/year. These two trends compare favorably with Aumman's night trend for 1231 cm$^{-1}$ of +2.9 ± 0.4 mK/year. It is interesting that our OISST trend differences agrees more closely with his RTGSST trend difference since these two data sets have similar heritage. Of course the extremely low statistical errors reported
by Aumann do not allow overlap of these two results, but that is not necessarily expected since we use different SST products. Agreement for AIRS radiometric trends at the several mK/year level for at least a single channel should be considered quite remarkable.

We also derived AIRS minus (ERA-I, OISST) SST trend differences using AIRS short wave only anomaly retrievals. For tropical latitudes, ± 30$^o$, the (AIRS - ERA-I) trend is 0.078 ± 0.040 K/decade and 0.065 ± 0.09 K/decade for OISST. These
615 represent significantly higher trend than observed using long and mid wave channels only. The trend difference between (AIRS

long wave minus AIRS short wave) anomaly fits is -0.058 ± 0.026 K/decade, clearly indicating the short wave positive drift relative to the long wave.

The latitude dependence of the AIRS derived SST trends versus ERA-I and OISST may eventually help determine the source of some of these differences. Figure 16 shows these trends between ± 60° latitude. The uncertainties in these trends are ~0.005K/year, but are not shown since these uncertainties are primarily geophysical in nature (how linear is the SST trend) and affect each SST product identically. Agreement is quite good among all products in the northern hemisphere, while OISST is systematically lower than AIRS and ERA-I in the southern hemisphere. Also shown are the AIRS SST trends using only the short wave channels (gray curve), which are always higher than the long wave AIRS trends except at the highest latitudes and near the equator.

Unfortunately, the AIRS Level 2 retrieval algorithm only uses short wave channels for surface temperature retrievals (Susskind et al., 2014). A recent inter-comparison of surface temperature trends from the AIRS Level 2 retrievals to three established surface temperature climate products (Susskind et al., 2019) concluded that the AIRS surface temperature trends were 0.24 K/decade, slightly higher than GISTEMP's (Hansen et al., 2010) value of 0.22 K/decade, and significantly higher than the HadCRUT4 (Morice et al., 2012) and Cowtan and Way (Cowtan et al., 2015) values of 0.17 and 0.19 K/decade respectively.

The results presented here conclude that the AIRS short wave channels are drifting positive by about 0.058 K/decade relative to the long wave channels, which appear to be in extremely good agreement with established SST climate products as discussed above. If we subtract this 0.058 K/decade AIRS short wave drift from the AIRS 0.24 K/decade trend presented in (Susskind et al., 2019) we obtain a corrected AIRS trend of 0.18 K/decade, much more in line with the HadCRUT4 and C+W values. In this case GISTEMP is now the only outlier. A more straightforward way to validate the reported AIRS Level 2 surface trends reported by (Susskind et al., 2019) would be to directly compare them to other SST products such as OISST, but unfortunately this was not part of the (Susskind et al., 2019) analysis.

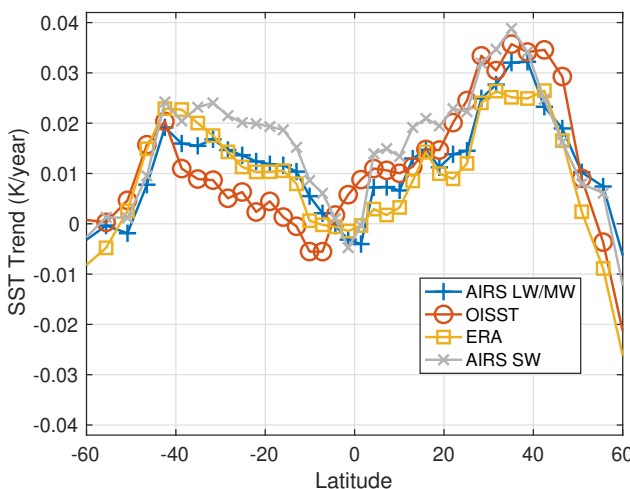

**Figure 16.** Latitude dependence of the linear trend in the AIRS retrieved SST, OISST, and ERA-I SST. Also shown are the SST trends when only the AIRS short wave channels are used to compute the anomalies.

## 5.8 CFC12 Retrieval

All anomaly retrievals presented here included CFC12 retrievals. Although these are not used for quantitative assessments of AIRS radiometric stability, the retrieved CFC12 anomaly is shown in Fig. 17 for completeness. Excellent agreement is found between the AIRS observed CFC12 and the ESRL Northern Hemisphere measurements (ESRL). The linear trends derived from these two curves are -2.94 ± 0.04 ppt/year for AIRS, and -2.93 ± 0.02 ppt/year for ESRL, nearly perfect agreement. These results give us confidence that the SST retrievals have not been compromised by CFC12 contamination, since there are a number of channels sensitive to both. Note that the trend of ~40 ppt of CFC12 derived here from AIRS is equivalent to only ~0.11K in BT!

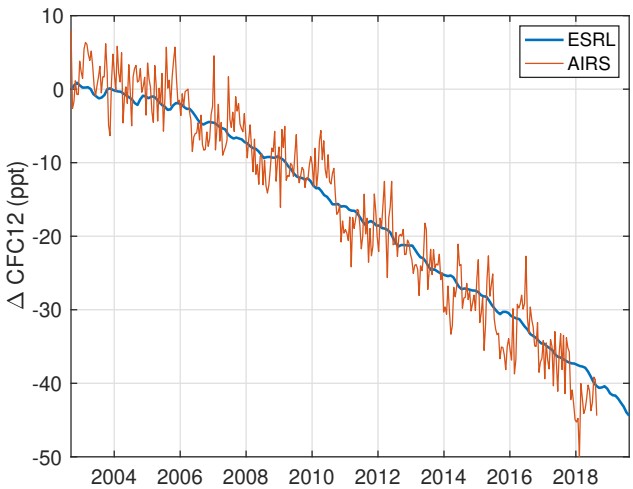

**Figure 17.** AIRS CFC12 retrieved anomaly compared to the NOAA ESRL Northern Hemisphere anomaly. Note that a 40 ppt trend in CFC12 corresponds to about 0.11K in brightness temperature for the channel with the highest CFC12 Jacobian.

## 6    Retrieval BT Breakouts and Residuals

The anomaly fit residuals provide a wealth of information on the behavior of each AIRS channel versus time. As stated earlier, unphysical shifts in the AIRS radiance time series can be reflected in either the retrieved geophysical anomalies or in the fit residuals. Jumps in the fit residuals will generally take place when the shifted radiances cannot be "adjusted away" by the BT Jacobians, which require a reasonably accurate physical response to radiance jumps. We believe that the anomaly retrieval approach presented here will allow objective corrections to AIRS radiances, especially for radiance jumps that can be tied to instrument events. The excellent agreement between the $CO_2$ and SST anomalies and in-situ data strongly suggests that the AIRS blackbody is very stable, which is key to climate-level trend measurements.

There are several likely causes for some of the differences seen here between our observed anomalies and the $N_2O$ and $CH_4$ truth anomalies from ESRL. Shifts in the frequency calibration of AIRS (Strow et al., 2006; Manning et al., 2019) have

655 largely been removed in the AIRS L1c product, although some transient shifts in the AIRS M-4a and M-4c arrays (that cover $N_2O$ and $CH_4$ channels) have not yet been corrected in L1c (see (Aumann et al., 2020)). The AIRS frequency shifts imply that detector views of the blackbody and cold scene targets have also shifted during the mission. While these shifts are very small, radiometric drifts/shifts could arise from these focal plane movements if the blackbody and cold scene targets are not perfectly uniform. As mentioned in Sec. 5.4, shifts of interference fringes in some of the AIRS entrance filters when Aqua was restarted

in Nov. 2003 may also contribute to the observed anomaly shifts. These fringe shifts have been modeled by the authors and future work may include modification of AIRS radiances before Nov. 2003 to remove the effects of these small shifts in the instrument spectral response function.

Here we present several views of the AIRS anomaly fits and their residuals as examples on how future work might proceed to potentially correct the AIRS radiances for small remaining radiometric drifts/shifts.

## 6.1 Retrieved Anomalies in BT Units

First to provide some context, Figure 18 shows the contribution of the various geophysical trends to the observed BT anomalies for channels sensitive to different geophysical variables. This is done by multiplying the BT Jacobian for some particular geophysical variable by its retrieved anomaly over time. For illustration purposed we averaged the trends over the latitude bins from $\pm 50°$latitude.

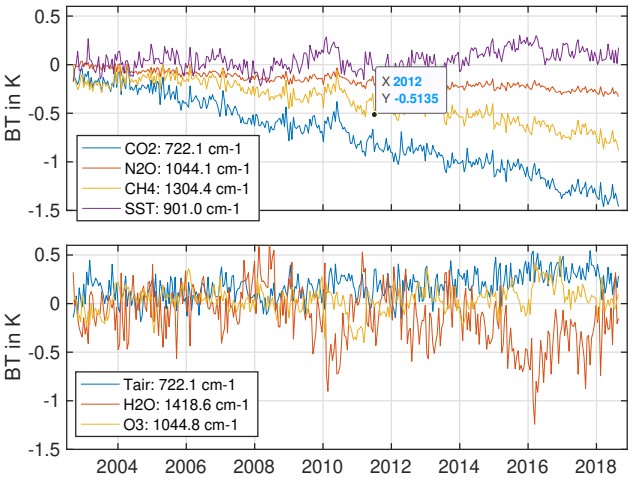

**Figure 18.** Contribution to the observed BT anomalies caused by the retrieved geophysical anomalies. These are simply the BT Jacobian multiplied by the time-dependent retrieved geophysical anomalies. The BT anomalies in the bottom panel are multiplied by the sum, over all layers, of the retrieved profile anomalies.

The upper panel shows that the retrieved $CO_2$ anomaly translates into a BT trend for the 722.1 cm$^{-1}$ channel of more than -1K. Channels very sensitive to the retrieved $CH_4$ and $N_2O$ anomalies have BT trends that are lower than $CO_2$. The anomaly for

a channel sensitive to SST in this panel has an upward trend due to increasing SST values, but these are quite small compared to the minor gas trends.

The bottom panel of Fig. 18 plots the BT anomalies due to the retrieved temperature, $H_2O$, and $O_3$ anomalies. The profile anomalies have been summed over all levels for this figure. The same channel chosen to illustrate the BT anomaly due to the $CO_2$ anomaly, 722.1 cm$^{-1}$, is also used to illustrate the contribution of the temperature anomaly. The BT trend for the 722.1 cm$^{-1}$ channel due to the temperature anomaly is far smaller than for $CO_2$, is slightly noisier, and has a small positive trend that mostly occurs after 2014. This would be expected since there is also a positive trend for SST with the same general time dependence. The BT trend due to the retrieved $H_2O$ anomaly is plotted for the 1418.6 cm$^{-1}$ channel sensitive to mid-tropospheric $H_2O$. This BT anomaly moves in the opposite direction to the BT anomaly due to temperature, which is expected since on a large scale increasing temperatures raise $H_2O$ amounts, which leads to lower BT values.

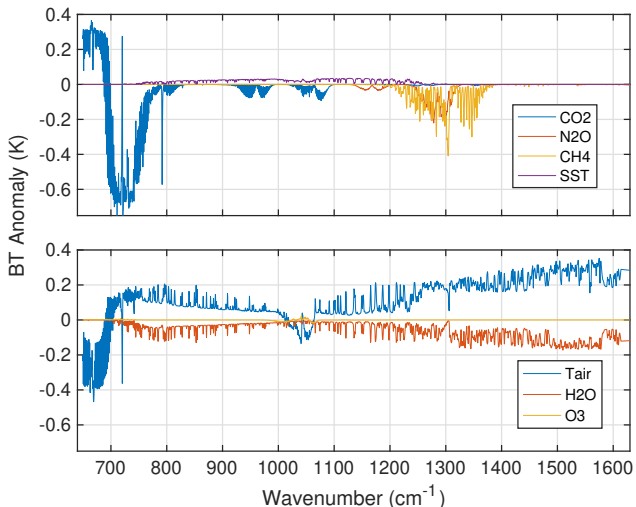

**Figure 19.** Contribution to the observed BT anomalies caused by the retrieved geophysical anomalies. These are simply the BT Jacobian multiplied by the mean, over time, of the 16-year record of geophysical anomalies. The BT anomalies in the bottom panel are multiplied by the sum, over all layers, of the retrieved profile anomalies.

Spectra illustrating how the various geophysical anomalies contribute to the BT anomalies are constructed by multiplying the BT Jacobians times the 16-year mean of the retrieved geophysical anomalies. Since these are computed quantities, all channels can be included. These are plotted in Fig. 19, where we separate the geophysical contributions just as in Fig. 18. If the trends are linear in time, the 16-year mean anomalies represent the anomalies for year eight. Dividing these by eight gives the nominal BT trend in K/year.

This figure clearly shows that $CO_2$ dominates the changes in most of the longwave region, as expected. The $N_2O$ and $CH_4$ BT anomalies are concentrated in the 1230-1400 cm$^{-1}$ region with significant overlap, which is largely separable in the retrieval. On this scale the BT changes due to SST are quite small. In the lower panel the temperature, $H_2O$, and $O_3$ BT anomaly trends are derived from the sum of the profile Jacobians over all layers. In many regions of the spectrum the temperature and $H_2O$

BT anomaly trends are dominant, an indication that our anomaly retrievals successfully accounted for variability in those parameters. BT trends in the channels sensitive to tropospheric temperature (700-750 cm$^{-1}$) are in the range of 0.01-0.02K/year (after dividing the plotted mean anomaly by eight), nominally consistent with global warming during this period.

The $H_2O$ greenhouse effect is clearly seen in the bottom panel of Fig. 19. The increased emission in the water band (1200-1615 cm$^{-1}$) due to higher atmospheric temperatures is largely negated by the decrease in emission due to increasing amounts of $H_2O$, which shifts the emission in any given channel to higher altitudes where the temperature is lower.

Also note that channels sensitive to stratospheric temperatures in the 650-690 cm$^{-1}$ region have a negative trend, indicating stratospheric cooling. This is also an expected result for global warming, but great care should be taken in using this data set for general conclusions since the sampling is non-uniform, and the air temperature trend standard deviation (Fig. 20) is about 80% larger than the air temperature trend shown in Fig. 19.

The uncertainties in the mean spectral BT anomalies shown in Fig. 19 can be estimated from the mean differences between the observed and computed BT anomalies per channel shown in Fig. 8. An average over all fitted channels gives a mean residual of -0.0021K ± 0.03K. This excellent fit, combined with the good agreement between the observed and in-situ truth data for the $CO_2$, $CH_4$, and $N_2O$ anomalies indicates that the anomalies shown in the top panel of Fig. 19 for a few sample channels are likely accurate to the anomaly fit 2-$\sigma$ uncertainty of level 0.03K.

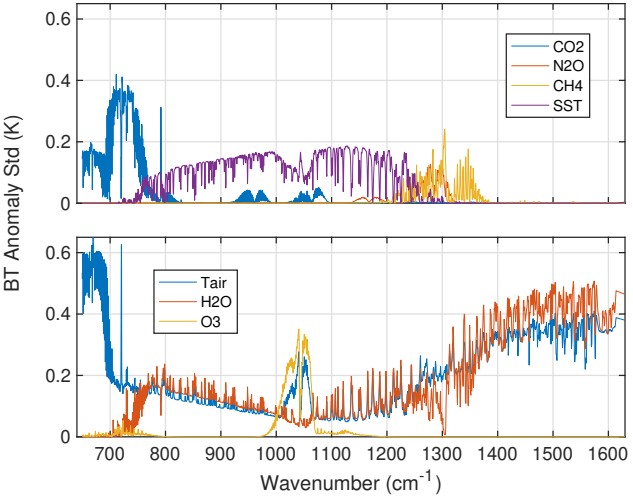

**Figure 20.** Standard deviation for the contribution to the observed BT anomalies caused by the mean retrieved geophysical anomalies shown in Fig. 19.

For completeness the standard deviation of the nominal linear anomaly trends shown in Fig. 19 are plotted in Fig. 20 using the same breakouts of geophysical anomalies. The $CO_2$ BT anomaly trend maximum standard deviation of ~0.35 K near 730 cm$^{-1}$ is nearly equal to the standard deviation expected if it was solely due to the linear trend in $CO_2$. The air temperature stratospheric standard deviation is large, as previously noted, presumably due to the effects of the quasi-biennial oscillation (QBO) and possibly ENSO variability. The variability due to air temperature and $H_2O$ produces standard deviations in the

water region (1250-1615 cm$^{-1}$) that are generally larger than variability due to trends in CH$_4$ and N$_2$O, but apparently our retrieval successfully removes those interferences. Note the relatively high O$_3$ variability, which we do retrieve but have not examined carefully. It is important to remember that these are anomaly standard deviations, so they do not include seasonal variability.

## 6.2 Anomaly BT Residuals

The anomaly fits shown above are summed and then subtracted from the observed BT anomalies to obtain the the BT anomaly fit residuals. Any trends in these residuals can also be examined to search for channels that changed characteristics during the 16-year time period.

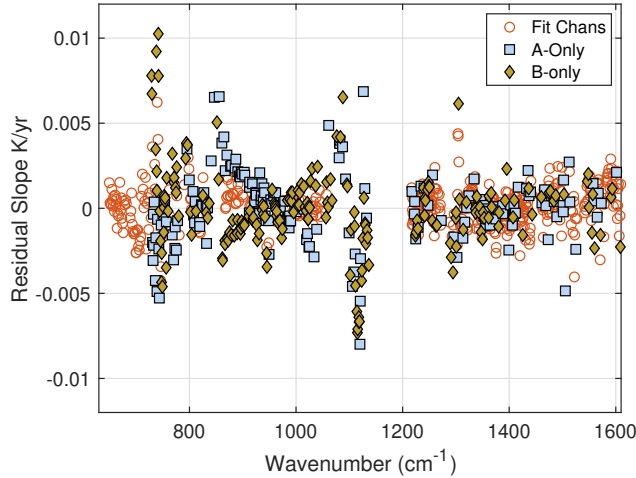

**Figure 21.** Slope of the AIRS anomaly residuals separated by A+B (Fit channels), A-Only, and B-Only. This illustrates trends in the A-only and B-only channels relative to A+B channels in some modules. The A-only and B-only channels were not used in the fitting, so they are not strictly residuals, but Observed - Computed differences.

Figure 21 shows the BT anomaly fit residual slopes for A+B, A-only, and B-only channels separately. Most of the A+B channels shown, all of which were used in the anomaly retrievals, are within ± 0.004 K/year of zero. While a large number of A-only and B-only channel are in agreement, there are a number of cases where they exhibit significant slopes (trends) that are not in agreement with the A+B channels. Module M-05 channels near 1100 cm$^{-1}$ are clearly drifting differently than the other channels (we did not use any A+B M-05 channels in the retrievals since they are also in error). Module M-08 channels near 851 cm$^{-1}$ show a clear separation between A+B channels and A-only, B-only. Clearly, the opposite signs of the A-only versus B-only drifts are largely cancelled when A+B channels are used. Since the SST retrievals are quite good, and because the surface channels near 1200 cm$^{-1}$ agree with the A+B channels, we conclude that the A-only and B-only drifts are real, and possibly due to drifts, or offsets, in the exact part of the blackbody and/or cold target scenes observed by these detectors.

Since the $N_2O$ retrieved anomalies exhibit some small unphysical behaviors, we examine the fit residuals for the 24 channels (used in the retrievals) that are most sensitive to $N_2O$. Visual inspection of these channel's residual time series clearly indicated

that 12 of them had easily identifiable features due to AIRS events. Figure 22 shows three different averages of these residual time series; (a) 12 good channels, with no strong evidence of AIRS events, (b) 12 bad channels which clearly exhibit jumps at the time of AIRS events, and (c) the mean time series for all 24 channels used in the anomaly fits. We see that the good channel mean (blue) is very flat, with a slight indication of a jump near the Nov. 2003 event. The bad channel curve (red) shows a large jump near Nov. 2003, possibly some longer-term drifts, and a feature in March 2014 that seems to last for 1 to 1 1/2 years.

This last feature can change sign depending on which bad channel is observed, making it very likely that this is due to the M-4a/M-4c frequency calibration shift that is not yet corrected in the L1c product.

A new set of anomaly retrievals were produced, but with the 12 bad $N_2O$ channels removed. When compared with the ESRL $N_2O$ anomalies, this change produced slightly better agreement with ESRL after Nov. 2013. The slope of the (AIRS - ESRL) anomaly difference curve was reduced from -0.141 K/Decade (as reported in Table 5) to -0.113 K/Decade, a slight

improvement. This drift relative to ESRL reduces to -0.069 K/Decade if anomaly data before Nov. 2013 is ignored. This illustrates that improvements to the AIRS products can be achieved by removing channels with residuals that have non-physical jumps. If the Nov. 2013 radiometric jumps can be removed (whether due to frequency shifts, fringe shifts, or pure radiometric jumps) even higher stability is possible. However, one could presently begin the AIRS time series, say on Jan. 1, 2004 and retain a stability approximately 2X better than climate trends.

These results illustrate a simple case for how the anomaly fit residuals can be used to improve AIRS trend products. In this work we have not looked for non-physical jumps in the retrieved temperature, $H_2O$, and $O_3$ profile anomalies. These products likely exhibit some of these behaviors and need to be included in any comprehensive study to further improve the AIRS radiance stability. Some sort of iterative approach will likely be needed in order to ensure that these small remaining radiometric jumps become undetectable in both the retrieved anomalies and in the anomaly residuals.

**7 Conclusions**

A framework for establishing stability of the AIRS radiances has been introduced that uses retrievals of minor gas and SST trends from BT anomaly spectra. Extremely good agreement between retrieved $CO_2$ trends (or anomalies) and in-situ trends from NOAA ESRL to -0.023 ± 0.009 K/Decade illustrates that a large fraction of AIRS channels are extremely stable, well below climate trends. The SST anomaly retrievals also compare favorably to the ERA-I reanalysis and to NOAA's OISST

SST product, with differences of less than 0.022 K/Decade, and slightly higher values for comparisons to OISST. Such good agreement for a wide range of detectors strongly suggests that the AIRS blackbody is very stable.

Unphysical radiometric jumps are observed in all the retrieved anomaly time series, but especially for $N_2O$ and $CH_4$. These jumps can largely be related to AIRS events, and we illustrate how the anomaly fit residuals, combined with inter-comparisons to truth anomaly trends such as $N_2O$, may provide a way to correct small remaining jumps in some AIRS channels.

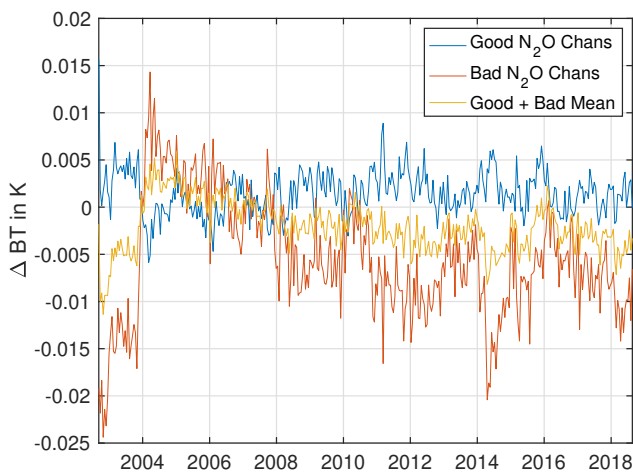

**Figure 22.** Anomaly fit residual time series for various combinations of 24 channels sensitive to $N_2O$ in the long wave. The bad $N_2O$ channels have easily visible jumps at times corresponding to AIRS hardware events.

This work emphasizes that users of AIRS radiances (both Level-1b and Level-1c) for climate applications must pay careful attention to channel selection, since certain detector arrays and channels are presently not suitable for climate trending, including all of the AIRS short wave channels. However, establishment of such a high level of stability for so many remote sensing observations/channels is unusual, and should lead to a high level of trust in AIRS climate trends that pay careful attention to only using validated climate-level channels.

**Acknowledgement**

The authors thank Steve Broberg, NASA JPL AIRS Project Office, for supplying us with a table of AIRS events. We also thank Steven Buczkowski at UMBC/JCET for the extensive data handling and production needed for this work. The hardware used for this work is part of the UMBC High Performance Computing Facility (HPCF). The facility is supported by the U.S. National Science Foundation through the MRI program (grant nos. CNS-0821258, CNS-1228778, and OAC-1726023) and the SCREMS program (grant no. DMS-0821311), with additional substantial support from the University of Maryland, Baltimore County (UMBC).

*Data availability.* channel list

**Appendix A: AIRS Detector Array Wavenumbers**

Table A1 shows the wavenumber ranges covered by each of the 17 AIRS arrays.

| Array Name | Start $\nu$ (cm$^{-1}$) | End $\nu$ (cm$^{-1}$) |
|:---:|:---:|:---:|
| 1a | 2552 | 2677 |
| 2a | 2432 | 2555 |
| 1b | 2309 | 2434 |
| 2b | 2169 | 2312 |
| 4a | 1540 | 1614 |
| 4b | 1460 | 1527 |
| 3 | 1337 | 1443 |
| 4c | 1283 | 1339 |
| 4d | 1216 | 1273 |
| 5 | 1055 | 1136 |
| 6 | 973 | 1046 |
| 7 | 910 | 974 |
| 8 | 851 | 904 |
| 9 | 788 | 852 |
| 10 | 727 | 782 |
| 11 | 687 | 729 |
| 12 | 649 | 682 |

**Table A1.** The wavenumber ranges covered by each of the 17 AIRS arrays.

## Appendix B: Anomaly and Profile Trend Retrievals

A complete simulated BT anomaly dataset was generated using ERA-I model fields, by matching each AIRS clear observation to ERA-I and generating a simulated radiance. This simulated dataset was used to set the regularization parameters for the profile inversions. The measurement of anomalies largely removes systematic errors in both the radiance observations (radiometric accuracy) and in the RTA (spectroscopy errors). We believe that these two factors helped make the retrieval inversions quite stable, requiring only minimal regularization.

Since our interest is mainly in the minor-gas profile offsets we used 20 atmospheric layers for the retrievals (20 each for temperature, $H_2O$, and $O_3$), created by concatenating layers from the 100-layer atmospheric profile model in (Strow et al., 2003). This choice, coupled with our regularization, provided more layers than degrees of freedom, as desired. We found that the low noise of the AIRS zonally averaged 16-day anomalies (see Sect. 4.1 coupled with low bias errors in the measurement covariances permitted the use of only minimal regularization.

Retrieval trials started with Tikhonov-only first-derivative (L1-type) regularization, which removes obvious outliers, mostly in the higher latitudes in the stratosphere. This gave averaged linear-trend accuracies in the simulations of -0.03 ± 0.07 K/year compared to the ERA-I model field trends used to generate the anomaly data set. (This degrades to -0.05 ± 0.08 K/year if

the regularization is lowered by a factor of 10X.) A reasonable goal is to achieve trends in simulation accurate to 0.01K/year,
averaged over the troposphere. A-priori uncertainties were then introduced for the temperature and $H_2O$ profiles of 2.5K and
60% respectively, which are roughly the maximum variation in these quantities over time for ± 50°latitude. These covariances
are not very restrictive given that measurement uncertainties are so low. It appears that their main impact is again for high
latitudes under conditions where we have higher noise due to low number of clear samples.

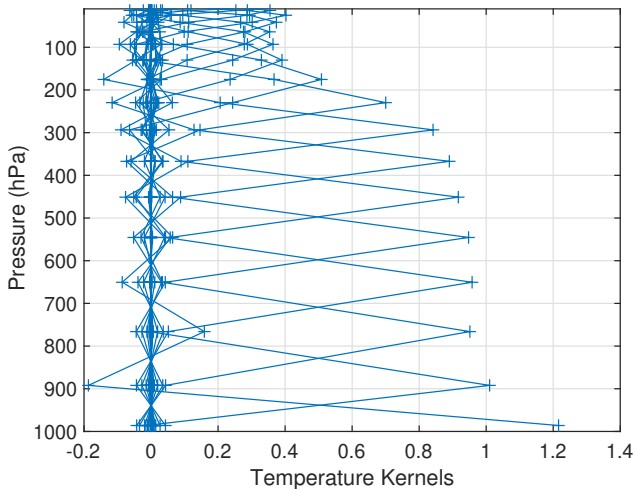

**Figure B1.** Temperature kernels for the anomaly retrievals. These are taken from a random day for the zonal bin centered at 28.3°N.

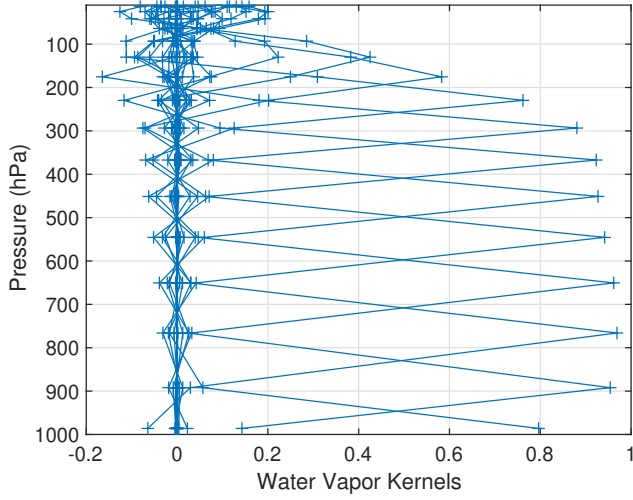

**Figure B2.** $H_2O$ kernels for the anomaly retrievals. These are taken from a random day for the zonal bin centered at 28.3°N.

The temperature and water vapor retrieval kernels are shown in Figs. B1,B2. They exhibit a very regular spacing in the
795 troposphere with roughly 12 well-separated kernels.

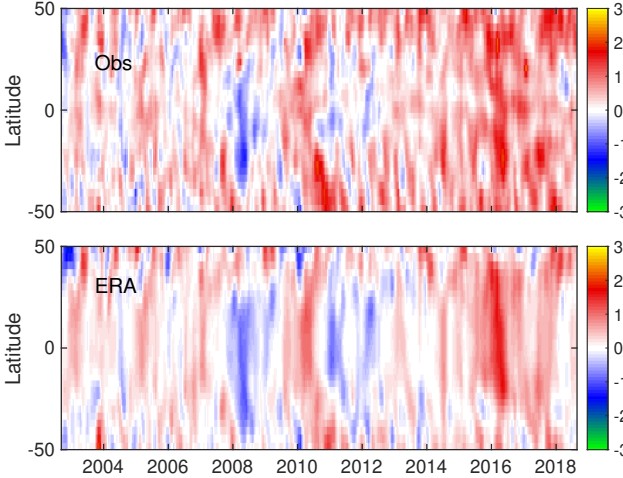

**Figure B3.** Retrieved 400 hPa temperature anomalies versus latitude. Top: Our retrievals from the AIRS observations. Bottom: ERA-I anomalies.

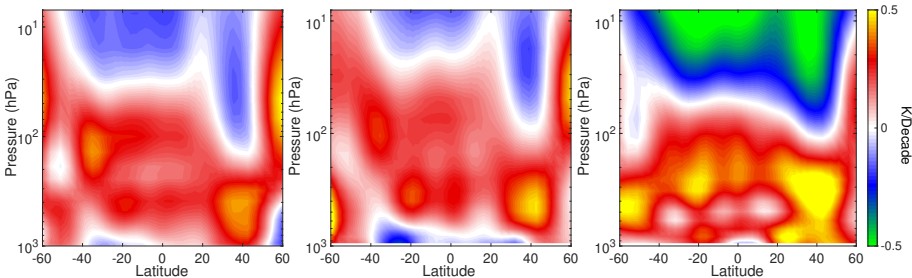

**Figure B4.** Temperature trends from the 16-year data period studied here. Left: ERA-I trends derived directly from the model temperature fields. Middle: Simulated retrievals of the ERA-I trends using radiance anomalies created from the ERA-I fields and our SARTA RTA. Right: Temperature profile trends retrieved from the AIRS observed anomalies. The middle panel simulation assumes that RTA is perfectly accurate.

Figure B3 illustrates the 400 hPa temperatures retrieved from the AIRS data (top panel) along with the ERA-I anomalies computed directly from the model fields. We do not expect these two data sets to compare perfectly, since for example, the ERA-I anomalies are from relatively large gridded data and the AIRS measurement are from a nominal 15 x 15 km field-of-view. Given the non-uniform sampling of this data set we do not think detailed examination of the observed versus ERA-I anomalies is warranted for scientific purposes. However, note that there are many similarities in time and latitude that give some measure of validation to our profile retrievals. Similar results are seen with water vapor profiles.

Figure B4 summaries the temperature trend simulations and comparisons between ERA-I trends, our anomaly retrievals from the ERA-I generated radiances, and those observed with the AIRS clear subset. The trends are computed from the anomaly retrievals (or model fields) using Eq. 6, where the input is the layer temperature instead of a $CO_2$ amount.

These results have been slightly smoothed to make visual inter-comparisons easier. The left panel shows the vertical trends versus latitude directly computed from the ERA-I temperature fields. The middle panel shows our simulated temperature trend retrievals. These simulations agree quite well with the ERA-I model fields, the largest differences are seen in the lower troposphere at the higher latitudes, and near the boundary layer in the tropics. The simulated retrievals are also placing the tropopause too high, not surprising given the lack of sensitivity of the infrared to the tropopause height and our limited number
of vertical layers. The right panel shows the temperature anomaly trends retrieved from the AIRS observed anomalies. Clearly there are significant differences between the ERA-I temperature profile trends and those we retrieved from AIRS, although the basic structure is relatively similar. Note that the uncertainties in these trends are quite high in the stratosphere (not shown) due to variations in the quasi-biennal oscillation (QBO), especially in the tropics, with errors larger than the observed trends in the vicinity of the tropopause. However, these uncertainties are largely present in both ERA-I and the AIRS observations.

The AIRS observed anomalies may also be impacted by errors in the BT Jacobians. The middle panel in Fig. B4 used similar RTAs for both simulations and the retrieval. The version of SARTA used for the radiance simulations is based on HITRAN2008 while the Jacobians used in the retrieval used kCARTA which is based on HITRAN2016 and a slightly modified version of $CO_2$ line-mixing. We expect that these spectroscopy differences have little impact since the $CO_2$ line strengths for the strong 15 $\mu$m bands have not changed between HITRAN versions. In addition, no noise was added to the simulated anomalies.

We believe that these results show that the anomaly retrievals used for measuring minor-gas trends exhibit realistic behavior and given our simulation testing this retrieval approach is likely to give accurate minor-gas trends. The impact of some of the regularization choices are discussed in Sect.5.4.

*Author contributions.* LLS led the study and made the comparisons between the anomaly fits and in-situ data. SDM developed the anomaly retrieval algorithm. LLS and SDM together optimized the anomaly retrieval regularization.

*Competing interests.* The authors declare that they have no conflict of interest.

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
