# Peer review of "Establishment of AIRS Climate-Level Radiometric Stability using Radiance Anomaly Retrievals of Minor Gases and SST"

_Atmospheric Measurement Techniques, 2019_

## Referee Comment (RC1) · Anonymous Referee #1 · 10 Mar 2020

This paper quantitatively evaluates the radiometric stability of AIRS observation. It provides an important guideline for future studies on climate-trend monitoring using AIRS and other infrared hyper-spectrometers. I believe it qualifies very well for this journal. It is well written and organized. I recommend this manuscript to be published after minor revision.

**Main Comments:**

[Figure]

Generally, I appreciate the logically organized approach present in this paper. Improvement can be made on the coherency of terms, explanation of figures, and other technical details. Further quantitative evaluation of this approach in the following aspects might be helpful:

1. Section 3.2  4.1: the Jacobian used in later retrieval could be sensitive to the temperature and water vapor amount, which is derived from the ERA-I dataset in the article [Line 217 to Line 223]. However, in Figure 2, besides a clear pattern in CO2 channels, the bias in O3 and H2O channels is large as well. This may imply biases in temperature/humidity profile in the ERA-I datasets, even for those channels clearly insensitive to the upper troposphere and stratosphere, which is not totally in agreement with the statement in Line 219 '...**ERA-I is so accurate we do not believe this is needed...**'.

I think it is important for the author to demonstrate, or at least mentioning in the text, whether Jacobian values of minor gases are sensitive to temperature and humidity, and whether updating them (besides gas amounts itself) is necessary.

2. Section 4.2: Can you clarify how Fig. 7 helps to evaluate the effect of Jacobian co-linearities?

3. Eq. 1: please define $r(t)$, $r_0$, and $\Phi$ in the text.

4. Can you add an equation to describe how a1 in Eq.1 and Eq. 2, and the directly retrieved quantity, x, is linked? In the Eq.1 and Eq.2, a1 terms are the linear trends of BT anomaly with time, but Fig. 11 and Line 340 treat it as the linear trend of individual gas amounts. I think it can be defined more carefully to avoid misunderstanding.

5. Section 5.7: Considering SST has a large diurnal fluctuation and a sun-synchronized orbit overpasses one geolocation approximately every 12 hours. Such temporal sampling may result in large bias in SST if directly compared it to a multi-day mean product. When compare AIRS retrieved SST and other products, have you considered the effect of this sampling difference?

6. It will be very interesting to see how the spectral anomaly at selected channels looks like and how it can be decomposed to spectral anomaly signal due to each retrieved anomaly (especially those discussed in the paper), compared to Fig.18.

Can you make a figure illustrating it? If possible, showing the standard deviation and linear trends of such spectral anomaly may be helpful to understand, besides the discussion showing in Fig.19, whether some channels are behaving no physically.

**Technical comments:**

1. Figure 2: . . . near 700-760 $cm^{-1}$ is due 'to' . . .

2. Line 161: delete extra 'by'.

3. Line 165: change 'influence' to 'influenced'.

4. Figure 5: . . . differences in the AIRS and ERA-I anomalies 'are'. . .

5. Line 248: change 'this' to 'these'.

6. Line 253: 'RTA' is never spelled out.

[Figure]

7. Line 266 to 267: considering rephrasing: ' because viewing angles to the Earth and cold scenes might change every so slightly'.

8. Line 294: delete 'in' located below . . .

9. Line 396: change 'use avoid . . .' to 'avoid using . . .' or consider rephrasing.

10. Line 412: change 'on' to 'one'.

11. Line 418: delete extra 'two'.

12. Line 450: change 'an' to 'a'.

13. Line 493: delete extra 'the'.

---

## Referee Comment (RC2) · Anonymous Referee #2 · 11 Mar 2020

The manuscript introduces and discusses important methodology and results to the suitability and utilisation of AIRS for climate applications, showing also a way for other hyperspectral sounder products (e.g. IASI, CrIS...). Based on a 16-year series, it indicates that AIRS radiance measurements and retrieved quantities match stability and sensitivity requirements for climate trend studies, as evaluated indirectly by Obs fit computations and direct intercomparisons to external reference measurements. This is found in line wit the scope of the journal and expected scientific novelty.

I find the manuscript overall very well structured and written, providing sufficient results

and discussions, with clear illustrations and appropriate references.

I recommend the publication of the manuscript pending few clarifications listed below.

—— General: "However, ERA-I is so accurate, that is not necessary" and similar other statement, sounds too absolute statement. The "so accurate" sould be elaborated a bit more, especially in view of some non-negligible biases seen in Fig.2.

L38: has it ever been considered to use AMSU in combination to disentangle T/CO2 signals? Like e.g. in Crevoisier et al. 2011 (TBC). would independence be more useful to climate studies, as oppoosed to using climatological CO2?

L56: not sure what the retrieval residuals can tell us really. The fit, if minimisation well programmed, will always come down to about the observation error in the end.

L65: how about any bias correction prior to the 1D-Var? NWP DA for instance need BC in variational minimisation to fit OBS with CALC. Has it been ever considered in AIRS L2 retrieval?

L115: over year+ ? clarify editorial

L161: by by (or bye bye typo)

L162: stddev in window may be due also to uncertainties in the forward modelling, including RTM/spectro as well as input SST/H2O profiles.

eq(5): explicit L?

L191: why are forward model uncertainties not included? The rationale (and consequences) should be discussed. Any bias correction?

L194: typo "more layers thAn"

L206: needs a little more explanation how the 0.004K and even 0.001K extremely low noise values were found. I assumed simple signal/noise enhancements resulting from massive averaging. However it is difficult to believe that one can fit the observation

down to that level, usually the RTM uncertainties combined with the effect of state vector not varied in the retrieval are larger than the instrument noise.

238, 264: incomplete ref (Aumann)

240-242: the DoF for O3 and H2O appears quite large compared to what is commonly accepted, as pointed out (usually $\sim$3 for O3 and 6-8 for H2O). I think this is more directly due to the massive averaging which effectively results in lowered instrument noise. 321 H20 channels on a single pixel would not bring such a high DoF, would it? Temeprature is a little under what is commonly expected of hyperspectral sounders $\sim$10-12 DoFs. But in this case, the channel pruning might be responsible for the signal loss.

254: typo to to (two to)

271: complete ref (Tans and Keeling)

L312: section reference broken

Fig.11: isn't it possible to plot break-down of ESRL components in their different latitudes location?

Explain Lag-1 autocorrelations

§5.5 For clarity, move Table 4 and Fig. 12 in section 5.5.

The larger departure AIRS - ESRL for CH4 and N2O over time is interesting, yet unexplained. Seems noticeable enough in Climate app context.

5.7: I understand that OSTIA provides the foundation SST (Merchant et al. 2014, Corlette et al., GHRSST website...), which is physically different to the radiative skin SST which is accessible to AIRS. In that respect, I find the agreement rather impressive with nearly no biases, while one could expect some given the different SST quantities. The authors should confirm the respective intrinsic nature of the SST datasets (model and retrieved) and possibly discuss the agreement accordingly. A correction of e.g.

skin-to-bulb bias of 0.17K may be necesssary in absolute term, it would however not impact the relative variation over time.

---

## Referee Comment (RC3) · Anonymous Referee #3 · 23 Mar 2020

Sergio DeSouza-Machado

**Anonymous Referee #3**

The authors present a novel approach with which to characterize decadal trends using AIRS radiance data. They apply this method to test instrument stability as well as temporal accuracy of retrieved geophysical variables over 16 years of AIRS measurements, limiting their scope to clear daytime scenes over ocean. With this work, the authors make a unique and valuable contribution to the science and application of satellite soundings.

This is a dense paper, and the authors expect the reader to hold on to an ever-increasing number of abstract concepts as the paper progresses. I suspect some of

the meaning and impact of their work may be lost as a result.

SCIENTIFIC ISSUES:

(1) Could the authors explain how they determine a scene to be over ocean? From Figure 1 it looks like coastlines are included.

(2) Determining clear scenes (Lines 91-93): The authors mention that the BT of each scene is subtracted from the BT of each of its 8 neighbors. Do the authors mean that they do this calculation for each 3 x 3 cluster of fields-of-view (i.e., within a field-of-regard), or do they treat each AIRS footprint (BT spectrum) independently and find 8 neighbors from adjacent fields-of-regard?

- What do the authors mean by "scene"? A field-of-view, or field-of-regard?

- My understanding here is that the authors select clear scenes based on two criteria, (i) scene uniformity, and (ii) accuracy of BT residuals, using ERA-I in simulation. This means that the authors select scenes for subsequent analyses only where ERA-I agrees well with the measured radiance. I feel one should keep this in mind when interpreting results. Could the authors clarify how may scenes are removed from each step?

- After applying these clear-sky filters, the authors then select ~20k scenes randomly. Given the total available, what percentage is this?

(3) Lines 106-107: This is the first time the authors introduce the AIRS Level 1c radiance product.

- Could the authors provide a reference here?

- What is the significance of using the L1c product?

- Do the authors use L1b radiances at all? If not, are the recommendations about radiometric stability and channel selection for the L1c product exclusively or does it also apply to L1b?
(4) Attributing results to sampling issues, Line 132 "the non-uniform spatial sampling", Line 162 "Some of this is likely due to changes in sampling from day to day", Line 166 "weather and sampling". I'm wondering how their sampling strategy could contribute large systematic effects in the results. If ∼20,000 scenes are randomly selected every day, then sampling variation from day to day will average out by design. The sampling bias should be a minimum. Could the authors elaborate on their reasoning here? I am wondering if some of the systematic effects visible in Figures 4 and 5 cannot not partly be explained by spectral interference from state variables used in simulation, especially those not present in ERA-I, like the minor gases.

(5) Line 168: "Note that since the ERA-I tracks the atmospheric state quite accurately most of the time-series "noise" is removed" Could the authors provide a reference here? How accurate is ERA-I compared to other models? Here, one should also remember that the authors specifically selected those scenes where ERA-I simulated BT spectral yielded a low residual. I feel that this simply demonstrates that their sampling strategy produced the desired results, not that ERA-I is accurate per se.

(6) Line 202: "These a-priori covariance uncertainty terms improved simulated retrievals and profile trends generated from these retrievals by 3-10%." Could the authors elaborate on this result? It appears like a large range and I'm wondering if improvements were limited to specific latitudinal zones or regions.

(7) Section 4.3 (Lines 250-259): This section is confusing to me. Could the authors better explain Figure 7? Is the panel on the left, "-55 deg latitude CO2 retrieval" for a single scene?

(8) Figure 9: How is it that the AIRS-ERA SST trend is a perfectly straight line across all wavenumbers?

(9) Lines 567-568: "This work emphasizes that users of AIRS data for climate applications must pay careful attention to channel selection since certain detector arrays and channels are presently not suitable for climate trending, including all of the AIRS short

wave channels"

- By "AIRS data", do the authors mean L1c?

- The authors demonstrated that they could calculate the shortwave spectral drift after the fact and, when subtracting it from the retrieved trend in Susskind et al. (2019), they could correct the trend sufficiently. Would such an approach not be a suitable alternative to channel selection? I imagine that the range of geophysical retrievals possible from bias-free channels must be reduced. This gives rise to the question whether climate-quality retrievals could be made from a reduction in spectral channels.

- How do the authors envisage the practical implementation of their recommendation here? The method the authors present here appears nuanced and expensive, not easy to implement by users of AIRS data.

- Do the authors consider publishing a list of AIRS channels suitable for climate applications?

TECHNICAL ISSUES:

- Discussion of spectral features: It will help the reader a great deal if the authors specify the wavenumber range they refer to with each mention of specific features, e.g., "CO2 region" in Line 160, or "upper-tropospheric water vapor" in Line 162, or "window region...water bands" in Line 207, etc.

- Line 28: "sea surface temperatures." Define the acronym "SST" upon first use.

- Line 59: "After a summarizing"

- Line 89: "Radiance (BT) anomalies" this is confusing since "BT" is an acronym or Brightness temperature, not Radiance.

- Line 106: "are matched to each clear scene are also saved"

- Figure 1 caption: "Density of AIRS clear ocean scene for calendar years 2012" should

be "Density of AIRS clear ocean scenes for calendar year 2012"

- Figure 2:

Caption: Should it not be "long-wave" instead of "short-wave"? Since the authors make specific reference to a CO2 feature, would they consider expanding the x-axis and adding more detailed tick marks to help the reader identify this feature specifically? As reader, I have the same issue with Figure 3 and its subsequent discussion.

Y-axis label: Reference to B(T) instead of BT. (Same in Fig. 4, Fig. 9)

- Line 161: "by by"

- Line 203: "each observations"

- Lines 202-203: Awkward sentence. Meaning unclear.

- Line 225: Add a comma to ease reading: "As discussed in Section 2, only"

- Lines 238, 264, 516: "(Aumann)" reference needs a date.

- Line 254: "to to"

- Figure 6 (page 11): legend should probably be "All channels used" for the blue profile?

- Line 267: "every so slightly"

- Line 294: "channels in located below 1615 cm-1"

- Line 300: Could the authors provide a reference for the L2c product here, so that the reader could follow up and better understand how "channels that do not exist . . .are filled during L1c creation".

- Line 312: "discussed in Sect. {sec:sst}"

- Line 316: "results presented here use avoid the short wave"

- Line 418: "just two two small"

- Line 546: "improvements to the AIRS products can be improved"

- Line 564: "jumps are observed in the all retrieved"
* * *

---

## Author Comment (AC1) · 29 May 2020

Authors responses to Atmos. Meas. Tech. Discuss., https://doi.org/10.5194/amt-2019-504

**Establishment of AIRS Climate-Level Radiometric Stability using Radiance Anomaly Retrievals of Minor Gases and SST**

by L. Larrabee Strow and Sergio DeSouza-Machado

Our responses are given below. For ease of review, we type-faced the reviewers questions in blue. At the bottom of our responses we include a copy of the new manuscript with removals in red and additions in blue with an underline. The main changes to the manuscript are:

1. Additional material on our clear selection has been added to Section 3 to show that we are not "matching" to ERA-I and indeed have objectively chosen the clear scenes.

2. Enhanced justification for using ERA-I model fields for evaluation of the anomaly retrieval Jacobians. This is done in a general manner in new Section 4.3. In Section 5 we include a detailed numerical analysis of any possible errors in our trend and anomaly results due to inaccurate ERA-I model fields, esp. see Table 4. (The blue underlining denoting new material in this table is overlapping the Jacobian denominators in Columns 1 and 2, but that will not show up in the final manuscript.)

3. The description of how we handle the co-linearity of the $CO_2$ and temperature Jacobians in Sect. 4.4 has been considerably lengthened. In addition, our main results shown in Table 3 now include (second row) an estimate of the AIRS stability using ESRL $CO_2$ trends as the standard peformed **without** applying our correction for the co-linearity of these Jacobians.

4. Section 6 now includes new material, as suggested by Reviewer 1, that illustrates how the AIRS BT anomaly trends are modified by individual gases, temperature, and water vapor.

5. In several places in this manuscript, such as Eq. 1 and 2, we have rewritten terms and tried to clarify the language so that the reader understands that our "measurement" is BT anomalies, not BT absolute spectra. This fact removes many concerns about biases in our data that seemed to trip people up.

**Reply to: Anonymous Referee 1**

Note: Many comments refer the reviewer to changes in the manuscript that are included at the end of this file.

*This paper quantitatively evaluates the radiometric stability of AIRS observation. It provides an important guideline for future studies on climate-trend monitoring using AIRS and other infrared hyper-spectrometers. I believe it qualifies very well for this journal. It is well written and organized. I recommend this manuscript to be published after minor revision.*

**1   Main Comments:**

*Generally, I appreciate the logically organized approach present in this paper. Improvement can be made on the coherency of terms, explanation of figures, and other technical details. Further quantitative evaluation of this approach in the following aspects might be helpful:*

We have tried to clarify a number of topics and terms in the text, based on this reviewers comments and the others.

**1.1   (1)**

*Section 3.2 4.1: the Jacobian used in later retrieval could be sensitive to the temperature and water vapor amount, which is derived from the ERA-I dataset in the article [Line 217 to Line 223]. However, in Figure 2, besides a clear pattern in CO2 channels, the bias in O3 and H2O channels is large as well. This may imply biases in temperature/humidity profile in the ERA-I datasets, even for those channels clearly insensitive to the upper troposphere and stratosphere, which is not totally in agreement with the statement in Line 219 "ERA-I is so accurate we do not believe this is needed". I think it is important for the author to demonstrate, or at least mentioning in the text, whether Jacobian values of minor gases are sensitive to temperature and humidity, and whether updating them (besides gas amounts itself) is necessary.*

Indeed, we just stated in the paper that ERA-Interim (ERA-I) was accurate enough. It is important to remember that the data being retrieved, BT anomalies, are very small, which implies that the Jacobians used in the retrieval do not have to be terribly accurate. This was also noticed by the other reviewers, therefore we put a significant effort into addressing the concerns about the accuracy of ERA-I for Jacobian evaluations.

Figure 2 has been enhanced to also show the single-footprint standard deviation between the AIRS observations and our ERA-I simulated BTs. We also include the AIRS noise in the new bottom panel of this figure, which shows that in the CO2 sounding region, the AIRS noise is barely smaller then the standard deviation of the biases, showing the ERA-I temperature fields are very close to what is observed.

A new section, 4.3 "Construction of Jacobians" has been added to address this issue qualitatively, introducing some estimates of ERA accuracy and AIRS radiometric accuracy (which would limit accuracy of standard retrievals used to generate the atmospheric state Jacobians, if we had done so.).

In Section 5.4 (CO2 anomalies) and 5.7 (SST anomalies) we now include a very detailed analysis of how potential uncertainties in the Jacobians generated using ERA-I would affect our anomaly retrievals and trends. These are summarized in a new Table (#4), which shows that using ERA-I for the Jacobians is likely to introduce errors that are far far below our statistical uncertainties. This is mainly due to the fact that the anomalies are quite small, so extremely accurate Jacobians are not needed.

I content that the ERA-I fields, once heavily averaged as in this paper, are more accurate than any 1D-var retrievals (like those done in the AIRS Level 2 product), since the assimilation includes many instruments, including AIRS, IASI, and CrIS.

**1.2   (2)**

*Section 4.2: Can you clarify how Fig. 7 helps to evaluate the effect of Jacobian co-linearities?*

We assume the reviewer is referring to Section 4.3 here, not 4.2.

Again, all the reviewers asked for a bit more clarification on how we removed the effects of co-linearity of the temperature and minor gases Jacobians on the retrievals. The real answer is that the internal consistency of our results show that our approach works extremely well. However, we have greatly expanded Section 4.4 (was section 4.3) to provide more context for our approach. I think our (new) quote from a TES paper by

Kulawik et. al. (2010) describes the usual approach, which is to use a-priori constraints to determine the partitioning of shared degrees of freedom between $CO_2$ and temperature". You have to do somethings, and instead of highly constraining the CO2 anomaly, we instead have good enough simulations that let us measure the effects of this co-linearity and remove it.

There was also a labelling error on the RHS plot in Fig. 7 that probably caused confusion and has been fixed.

Moreover, we added an entry to Table 3 (new second row) that shows the trend differences between AIRS and ESRL if we do NOT do this co-linearity correction. It makes little difference to the mean trend difference (changes the sign, but with and without this correction the difference are very small). However, the uncertainty in the trend difference is almost 3X higher without the co-linearity correction.

**1.3 (3)**

Eq. 1: please define r(t), r0 , and $\phi$ in the text.

Sorry, all fixed. We also neglected to explicitly define **L** the full regularization operator in Eq.2, which is now fixed as well.

**1.4 (4)**

Can you add an equation to describe how a1 in Eq.1 and Eq. 2, and the directly retrieved quantity, x, is linked? In the Eq.1 and Eq.2, a1 terms are the linear trends of BT anomaly with time, but Fig. 11 and Line 340 treat it as the linear trend of individual gas amounts. I think it can be defined more carefully to avoid misunderstanding.

We have de-emphasized the BT linear rate term $a_1$ by re-writing Eq. 2 in a more standard form. $a_1$ was only inserted into Eq. 1 as a diagnostic (shown in Fig. 3) and is NOT used in the anomaly retrievals. In the vicinity of Line 340 we have restated our fitting function used for the geophysical anomaly fits, and now use the term $b_1$ for the geophysical rate instead of $a_1$ used to determine the radiance/BT linear rates.

**1.5 (5)**

Section 5.7: Considering SST has a large diurnal fluctuation and a sunsynchronized orbit overpasses one geolocation approximately every 12 hours. Such temporal sampling may result in large bias in SST if directly compared it to a multi-day mean product. When compare AIRS retrieved SST and other products, have you considered the effect of this sampling difference?

Indeed our biases relative to ERA-I for SST will likely include a diurnal component as our clear subsets change location in space and time. Note that (I think) ERA-I uses a single SST per day (ECMWF forecast tries to add in diurnal, but getting the details from ECMWF is hard.) But, this doesn't matter for the trends, and matters very little for the SST anomaly time series since most of the time sampling variability averages out. For the 16-year trends it is a total non-issue since our sampling (averaged over days to months) is extremely stable in time.

We quantify this in Section 3.1. We looked at the sampling time trends (which we had never done before) and found that the mean trend in time sampling over the 16-years was -20 ± 40 **seconds**. Amost nothing. There is a sampling trend that varies slightly with season, but that is small and would be removed when forming the anomalies anyway.

**1.6 (6)**

It will be very interesting to see how the spectral anomaly at selected channels looks like and how it can be decomposed to spectral anomaly signal due to each retrieved anomaly (especially those discussed in the paper), compared to Fig.18.

Can you make a figure illustrating it? If possible, showing the standard deviation and linear trends of such spectral anomaly may be helpful to understand, besides the discussion showing in Fig.19, whether some channels are behaving no physically.

We thought this was a good idea, mostly we didn't do it to keep the paper as short as possible. We have now added three figures and some discussion on this topic in new Section 6.1. I think it does help the reader get some context on what we did. Thanks.

**2   Technical comments:**

All comments listed below have been addressed.

1. Figure 2: ... near 700-760 cm-1 is due 'to' ...
2. Line 161: delete extra 'by'.
3. Line 165: change 'influence' to 'influenced'.
4. Figure 5: ... differences in the AIRS and ERA-I anomalies 'are' Printer-friendly version
5. Line 248: change 'this' to 'these'.
6. Line 253: 'RTA' is never spelled out.
7. Line 266 to 267: considering rephrasing: ' because viewing angles to the Earth and cold scenes might change every so slightly'.
8. Line 294: delete 'in' located below ...
9. Line 396: change 'use avoid ...' to 'avoid using ...' or consider rephrasing.
10. Line 412: change 'on' to 'one'.
11. Line 418: delete extra 'two'.
12. Line 450: change 'an' to 'a'.
13. Line 493: delete extra 'the'.

[revised manuscript text omitted]

---

## Author Comment (AC2) · 29 May 2020

**Authors responses to Atmos. Meas. Tech. Discuss., https://doi.org/10.5194/amt-2019-504**

**Establishment of AIRS Climate-Level Radiometric Stability using Radiance Anomaly Retrievals of Minor Gases and SST**

**by L. Larrabee Strow and Sergio DeSouza-Machado**

Our responses are given below. For ease of review, we type-faced the reviewers questions in blue. At the bottom of our responses we include a copy of the new manuscript with removals in red and additions in blue with an underline. The main changes to the manuscript are:

- 1. Additional material on our clear selection has been added to Section 3 to show that we are not "matching" to ERA-I and indeed have objectively chosen the clear scenes.
- 2. Enhanced justification for using ERA-I model fields for evaluation of the anomaly retrieval Jacobians. This is done in a general manner in new Section 4.3. In Section 5 we include a detailed numerical analysis of any possible errors in our trend and anomaly results due to inaccurate ERA-I model fields, esp. see Table 4. (The blue underlining denoting new material in this table is overlapping the Jacobian denominators in Columns 1 and 2, but that will not show up in the final manuscript.)
- 3. The description of how we handle the co-linearity of the CO2 and temperature Jacobians in Sect. 4.4 has been considerably lengthened. In addition, our main results shown in Table 3 now include (second row) an estimate of the AIRS stability using ESRL CO2 trends as the standard peformed **without** applying our correction for the co-linearity of these Jacobians.
- 4. Section 6 now includes new material, as suggested by Reviewer 1, that illustrates how the AIRS BT anomaly trends are modified by individual gases, temperature, and water vapor.
- 5. In several places in this manuscript, such as Eq. 1 and 2, we have rewritten terms and tried to clarify the language so that the reader understands that our "measurement" is BT anomalies, not BT absolute spectra. This fact removes many concerns about biases in our data that seemed to trip people up.

**Reply to: Anonymous Referee 2**

Note: Many comments refer the reviewer to changes in the manuscript that are included at the end of this file.

**1** General Comments**

The manuscript introduces and discusses important methodology and results to the suitability and utilisation of AIRS for climate applications, showing also a way for other hyperspectral sounder products (e.g. IASI, CrIS...). Based on a 16-year series, it indicates that AIRS radiance measurements and retrieved quantities match stability and sensitivity requirements for climate trend studies, as evaluated indirectly by Obs fit computations and direct intercomparisons to external reference measurements. This is found in line wit the scope of the journal and expected scientific novelty.

I find the manuscript overall very well structured and written, providing sufficient results and discussions, with clear illustrations and appropriate references.

I recommend the publication of the manuscript pending few clarifications listed below.

**2 Use of ERA for Jacobians**

"However, ERA-I is so accurate, that is not necessary" and similar other statement, sounds too absolute statement. The "so accurate" sould be elaborated a bit more, especially in view of some non-negligible biases seen in Fig.2.

Indeed, we just stated in the paper that ERA-Interim (ERA-I) was accurate enough. It is important to remember that the data being retrieved, BT anomalies, are very small, which implies that the Jacobians used in the retrieval do not have to be terribly accurate. This was also noticed by the other reviewers, therefore we put a significant effort into addressing the concerns about the accuracy of ERA-I for Jacobian evaluations.

The biases in Fig. 2 are actually not that big. Note that we are showing a slightly different bias in the revised paper. Since our RTA (SARTA) has a default CO2 amount of 385 ppm we selected a time period where the atmosphere had the same CO2 amount as well. That made the biases in the CO2 sensitive region from 700 to 750 cm-1 even smaller, around 0.2-0.3K on average. The AIRS radiometry may not be that accurate. The water region biases (1300-1615 cm-1) are larger, but we do not heavily rely on accurate water vapor in the mid-troposphere for this work. Even if the Jacobians valuels are slightly incorrect, our retrieval still removes their effect in terms of interference with  $N_{2}$  and  $CH_{4}$ . We are somewhat sensitive to the column amount of water vapor for window region channels that are used to fit for the SST anomalies. However, as noted below, we added quite a bit of material to justify the use of ERA for Jacobian evaluation and find that this introduces extremely small inaccuracies that can be ignored. Some details of what we did are discussed below.

Figure 2 has been enhanced to also show the single-footprint standard deviation between the AIRS observations and our ERA-I simulated BTs. We also include the AIRS noise in the new bottom panel of this figure, which shows that in the CO2 sounding region, the AIRS noise is barely smaller then the standard deviation of the biases, showing the ERA-I temperature fields are very close to what is observed.

A new section, 4.3 "Construction of Jacobians" has been added to address this issue qualitatively, introducing some estimates of ERA accuracy and AIRS radiometric accuracy (which would limit accuracy of standard retrievals used to generate the atmospheric state Jacobians, if we had done so.).

In Section 5.4 (CO2 anomalies) and 5.7 (SST anomalies) we now include a very detailed analysis of how potential uncertainties in the Jacobians generated using ERA-I would affect our anomaly retrievals and trends. These are summarized in a new Table (#4), which shows that using ERA-I for the Jacobians is likely to introduce errors that are far far below our statistical uncertainties. This is mainly due to the fact that the anomalies are quite small, so extremely accurate Jacobians are not needed.

I content that the ERA-I fields, once heavily averaged as in this paper, are more accurate than any 1D-var retrievals (like those done in the AIRS Level 2 product), since the assimilation includes many instruments, including AIRS, IASI, and CrIS.

**3** Specific Comments**

• L38: has it ever been considered to use AMSU in combination to disentangle T/CO2 signals? Like e.g. in Crevoisier et al. 2011 (TBC). would independence be more useful to climate studies, as oppoosed to

**using climatological CO2?**

This is certainly a valid idea and many people use it. We did not want to invoke another instrument in these anomaly retrievals since that introduces uncertainties in say, the AMSU radiometric stability, which could greatly complicate the analysis. Our temperature trends are quite similar to those in ERA, which lends credibility that we have indeed separated  $CO_2$  from temperature. In addition, we added quite a bit of material to the new Section 4.4 to justify our approach on separating  $CO_2$  from temperature. We believe the results shown show that this was successful. Please note that the RHS plot in Fig. 7 had a labelling error that may have introduced some confusion on our mitigation of co-linearities in the  $CO_2$  and temperature Jacobians.

• L56: not sure what the retrieval residuals can tell us really. The fit, if minimisation well programmed, will always come down to about the observation error in the end.

Exactly. And what this paper is after **is** the observation error, ie how stable is AIRS, what channels are mis-behaving and by how much. This is discussed in Section 6.2.

• L65: how about any bias correction prior to the 1D-Var? NWP DA for instance need BC in variational minimisation to fit OBS with CALC. Has it been ever considered in AIRS L2 retrieval?

That is the whole point of retrieving from BT anomalies. AIRS calibration errors (that do not change in time) are removed when forming the BT anomaly. And, that also means we only use relative changes in the RTA to determine the anomalies, so RTA bias is removed as well. That is why this approach works so well. In order to minimize confusion on this point we have added a sentence at the end of Sect. 4.1 stating this more explicitly, and have changed Eqs. 1 and 2 slightly to make it a bit more transparent that we are retrieving from a BT anomaly. In addition, we also clarified our approach by explicitly defining the observation y in Eqs. 4 and 5 in terms of the BT anomaly derived in Eq. 2.

• L115: over year+? clarify editorial

We changed the working to "mult-year".

• L161: by by (or bye bye typo)

Fixed.

• L162: stddev in window may be due also to uncertainties in the forward modelling, including RTM/spectro as well as input SST/H2O profiles. eq(5): explicit L?

Yes, we didn't explicitly define *L*, now fixed. The stddev here would be extremely insensitive to forward modelling errors and spectroscopy because these are mean anomalies, and only represent small changes in the atmospheric state, not how well we can simulate the observed BT. Agreed that SST/H2O profile errors could cause some of the window STD, now stated in the paper.

• L191: why are forward model uncertainties not included? The rationale (and consequences) should be discussed. Any bias correction?

There is no concept of a bias correction here, since we are retrieving anomalies. Time independent bias errors in the AIRS calibration are removed in forming the anomaly, and the retrieval of anomalies does not need to compute the absolute BT, only it's variations. The forward model is fixed throughout this time period so there was no need to put in any RTA uncertainties into the retrieval. There **are** second order RTA errors that come into the retrieval via the Jacobians, and these are discussed in detail in the revised paper.

Please note that our a-priori values for the temperature and water profiles is "zero", as well as our linearization point. That's because these are anomalies, not absolute BTs. Granted, we need the atmospheric state for computing the Jacobians, but ERA-I serves that purpose with high accuracy for our purposes.

• L194: typo "more layers thAn"

Fixed.

• L206: needs a little more explanation how the 0.004K and even 0.001K extremely low noise values were found. I assumed simple signal/noise enhancements resulting from massive averaging. However it is difficult to believe that one can fit the observation down to that level, usually the RTM uncertainties combined with the effect of state vector not varied in the retrieval are larger than the instrument noise.

We tried to clarify this, but it is indeed due to massive averaging. I agree, it was hard for us to believe we needed to faithfully use these low errors in our OE retrieval, but as stated in the paper, until we did that we could not retrieve the large  $CO_2$  anomalies near the end of the time series. Again, we are retrieving BT anomalies, and they are very small (not much larger than single footprint noise). And, as stated above, using anomalies avoids calibration and RTA bias errors, resulting in very robust retrievals.

• 238, 264: incomplete ref (Aumann)

Fixed.

• 240-242: the DoF for O3 and H2O appears quite large compared to what is commonly accepted, as pointed out (usually  $\approx$  3 for O3 and 6-8 for H2O). I think this is more directly due to the massive averaging which effectively results in lowered instrument noise. 321 H2O channels on a single pixel would not bring such a high DoF, would it? Temeprature is a little under what is commonly expected of hyperspectral sounders  $\approx$  10-12 DoFs. But in this case, the channel pruning might be responsible for the signal loss.

Yes, we agree. It's the low noise that does this. Generally you would expect 5 DOFS or so for H2O. Indeed the high number of  $H_2O$  channels contributed to this DOF, and we ignored all sorts of correlations that might exist. We didn't pursue this in detail because it is not all that relevant for this study, where we are not that interested in the final temperature and  $H_2O$  profiles, as long as their trends are reasonable. We also used a very simple approach for the DOF measurements, using only the mean profile. I strongly suspect that that these averaged anomalies, and associated trends don' have enough variability to need more than a few DOFs. These issues will be pursued in another study where we are not concentrating just on highly averaged clear scene anomalies.

• 254: typo to to (two to)

Fixed.

• 271: complete ref (Tans and Keeling)

I am doing what their web site suggests, and they don't give a date. Maybe the AMT editors/proofreaders can help me on this.

• L312: section reference broken

Fixed.

• Fig.11: isn't it possible to plot break-down of ESRL components in their different latitudes location?

That would involve using a much more complicated ESRL data set that is not gridded by latitude, but just provides point sources. Or, it would involve using CarbonTracker, which is an assimilated product that I though was not appropriate. What ESRL does provide in simple form are a few high-quality stations (MLO and CGRIM) and a "global"  $CO_2$  product which is easy to use. Since we are after AIRS stability in this paper, using the "global" product seemed appropriate. A scientific study of  $CO_2$  spatial variability using this appproach would need more complete in-situ  $CO_2$  data, but that is for another study.

• Explain Lag-1 autocorrelations

We have removed several reference to lag-1 correlations in Section 3.3 since they are not needed at that point. Later in the paper, when we do discuss "lag-1 autocorrelations" we clarify that these are corrections to least-squares uncertainty estimates for the serial correlations in the anomaly time series.

The Santer reference is very widely used in climate research and it details how the lag-1 autocorrelation of a time series can be used to empirically correct least-squares uncertainty estimates that assume that the time series (residuals) will have uncorrelated gaussian noise (which is rarely true). An explanation of the lag-1 approach would be too detailed for this paper and should be unnecessary given its popularity in the climate community.

• 5.5 For clarity, move Table 4 and Fig. 12 in section 5.5.

I can't do that with latex submissions. Hopefully the AMT typesetting will fix that problem.

• The larger departure AIRS - ESRL for CH4 and N2O over time is interesting, yet unexplained. Seems noticeable enough in Climate app context.

I do not think that departure is correct, it is due to some shifts in the AIRS calibration due to AIRS hardware "events". See the discussion in Section 5.5 for the details on this.

• 5.7: I understand that OSTIA provides the foundation SST (Merchant et al. 2014, Corlette et al., GHRSST website...), which is physically different to the radiative skin SST which is accessible to AIRS. In that respect, I find the agreement rather impressive with nearly no biases, while one could expect some given the different SST quantities. The authors should confirm the respective intrinsic nature of the SST datasets (model and retrieved) and possibly discuss the agreement accordingly. A correction of e.g. skin-to-bulb bias of 0.17K may be necessary in absolute term, it would however not impact the relative variation over time.

We are only measuring OSTIA (via ERA-I) and OISST SST trends and anomallies. So, any biases in these products relative to AIRS radiances would not show up in our analysis.

[revised manuscript text omitted]

---

## Author Comment (AC3) · 29 May 2020

Authors responses to Atmos. Meas. Tech. Discuss., https://doi.org/10.5194/amt-2019-504

**Establishment of AIRS Climate-Level Radiometric Stability using Radiance Anomaly Retrievals of Minor Gases and SST**

by L. Larrabee Strow and Sergio DeSouza-Machado

    Our responses are given below. For ease of review, we type-faced the reviewers questions in blue. At the bottom of our responses we include a copy of the new manuscript with removals in red and additions in blue with an underline. The main changes to the manuscript are:

1. Additional material on our clear selection has been added to Section 3 to show that we are not "matching" to ERA-I and indeed have objectively chosen the clear scenes.

2. Enhanced justification for using ERA-I model fields for evaluation of the anomaly retrieval Jacobians. This is done in a general manner in new Section 4.3. In Section 5 we include a detailed numerical analysis of any possible errors in our trend and anomaly results due to inaccurate ERA-I model fields, esp. see Table 4. (The blue underlining denoting new material in this table is overlapping the Jacobian denominators in Columns 1 and 2, but that will not show up in the final manuscript.)

3. The description of how we handle the co-linearity of the $CO_2$ and temperature Jacobians in Sect. 4.4 has been considerably lengthened. In addition, our main results shown in Table 3 now include (second row) an estimate of the AIRS stability using ESRL $CO_2$ trends as the standard peformed **without** applying our correction for the co-linearity of these Jacobians.

4. Section 6 now includes new material, as suggested by Reviewer 1, that illustrates how the AIRS BT anomaly trends are modified by individual gases, temperature, and water vapor.

5. In several places in this manuscript, such as Eq. 1 and 2, we have rewritten terms and tried to clarify the language so that the reader understands that our "measurement" is BT anomalies, not BT absolute spectra. This fact removes many concerns about biases in our data that seemed to trip people up.

**Reply to: Anonymous Referee 3**

Note: Many comments refer the reviewer to changes in the manuscript that are included at the end of this file.

**1 General**

The authors present a novel approach with which to characterize decadal trends using AIRS radiance data. They apply this method to test instrument stability as well as temporal accuracy of retrieved geophysical variables over 16 years of AIRS measurements, limiting their scope to clear daytime scenes over ocean. With this work, the authors make a unique and valuable contribution to the science and application of satellite soundings.

This is a dense paper, and the authors expect the reader to hold on to an everincreasing number of abstract concepts as the paper progresses. I suspect some of the meaning and impact of their work may be lost as a result.

The main issue appears to be that the concept of retrieving atmospheric state anomalies directly from the observed BT spectra anomalies is not always kept in mind, and concerns that are raised are only relevant for standard state retrievals from absolute BT spectra. We have tried to clarify what is being retrieved in several places in the manuscript, including changes to Eq. 2. We also added a sentence to the end of Sect. 4.1 to emphasize we are retrieving anomalies from **BT anomalies**.

**2 Scientific Issues**

**2.1 (1)**

Could the authors explain how they determine a scene to be over ocean? From Figure 1 it looks like coastlines are included.

Coastlines are not included. The AIRS Level 1 landfrac value must equal 0 for us to call the scene "ocean".

**2.2 (2)**

Determining clear scenes (Lines 91-93): The authors mention that the BT of each scene is subtracted from the BT of each of its 8 neighbors. Do the authors mean that they do this calculation for each 3 x 3 cluster of fields-of-view (i.e., within a field-of regard), or do they treat each AIRS footprint (BT spectrum) independently and find 8 neighbors from adjacent fields-of-regard?

We treat each AIRS footprint independently and find 8 neighbors from adjacent "fields-of-regard". The concept of "field-of-regard" does not really exist in the level 1 data (the term does not appear in the Level 1b ATBD). I see no reason to introduce a constract of the AIRS Level 2 retrieval that has no meaning for this work.

- What do the authors mean by "scene"? A field-of-view, or field-of-regard?

In Section 3.1 we added a sentence to clarify more technically what we mean by scene. It is a single footprint. Again the concept of field-of-regard plays no role in this work.

- My understanding here is that the authors select clear scenes based on two criteria, (i) scene uniformity, and (ii) accuracy of BT residuals, using ERA-I in simulation. This means that the authors select scenes for subsequent analyses only where ERA-I agrees well with the measured radiance. I feel one should keep this in mind when interpreting results. Could the authors clarify how may scenes are removed from each step?

This is a far too general statement. We only compare ERA-I simulated radiances to the measured radiances for two window channels, as stated in the text. No other channels are compared, since we are looking for cloud contamination.

We have added more information on this process as requested in Section 3.1 where we now state the the 4K ERA-I bias test removes ~20% of the scenes detected with the uniformity filter. A map of these scenes shows very clearly that they are due to marine boundary layer stratus clouds, which form along the west coasts of the Americas and Africa.

Moreover, we now state the the distribution of observed biases for the very clear 1231 cm-1 channel is almost gaussian with a width of ~0.6K. The wing of this distribution is near zero for the -4K cutoff used for marine boundary layer stratus.

So, there is little "matching" to ERA-I at all. This is emphasized by changing Fig. 2 to include the single footprint standard deviation of the bias between ERA-I simulations and AIRS observations. We show in the figure that in the $CO_2$ region from 700-750 cm$^{-1}$ that the ERA-I bias standard deviation is just every so slightly higher than the AIRS NEDT, consequently the ERA-I model fields for temperature follow the AIRS observations very closely.

- After applying these clear-sky filters, the authors then select  20k scenes randomly.  Given the total available, what percentage is this?

We mistakenly stated 20K scenes were selected, our hard limit was upped to 40K a while ago. Now, that hard limit is almost never reached. Since we are using descending only, our total number of clear scenes is in the 10K range for most days. This is now stated in the paper.

**2.3  (3)**

Lines 106-107: This is the first time the authors introduce the AIRS Level 1c radiance product.

- Could the authors provide a reference here?

Done, and now with the date.

- What is the significance of using the L1c product?

We added 2 paragraphs in Sect. 3.1 that covers this.

- Do the authors use L1b radiances at all? If not, are the recommendations about radiometric stability and channel selection for the L1c product exclusively or does it also apply to L1b?

We added some text that basically says there is almost no difference between L1b and L1c for the "real" channels used i our retrievals, except for quite small adjustment in L1c for drifts in the AIRS frequency scale.

**2.4  (4)**

Attributing results to sampling issues, Line 132 "the non-uniform spatial sampling", Line 162 "Some of this is likely due to changes in sampling from day to day", Line 166 "weather and sampling". I'm wondering how their sampling strategy could contribute large systematic effects in the results. If  20,000 scenes are randomly selected every day, then sampling variation from day to day will average out by design. The sampling bias should be a minimum. Could the authors elaborate on their reasoning here? I am wondering if some of the systematic effects visible in Figures 4 and 5 cannot not partly be explained by spectral interference from state variables used in simulation, especially those not present in ERA-I, like the minor gases.

I disagree. Regions of clear scenes vary from day-to-day, especially in times of ENSO events. The clear sampling is far from uniform!

Also, I think the reviewer is thinking of the data in Fig. 4 as a bias. It is not, it is the BT **anomaly**. So, for example, the value of ~0.25K in the window region in Fig. 4, blue curve, means that that channel "changed" during the time period by ~0.25K/8 years, since the anomaly is 16-years long, the mean is representative of the average anomaly after 8 years. Divide 0.25K/8year and you get 0.03K/year, which is very close to the ERA-I SST rate of increase for this latitude, over the 16 years. This is not a bias plot! And the blue curve is not showing a systematic "effect" other than climate change.

Figure 5 just says that for this channel (710.14 cm$^{-1}$, now specified in the figure caption) that the radiance in the 28.3° North latitude bin varies by ~0.5K day to day. That is extremely small for a non-uniform sampling over longitude! The "Noise" in the blue and red curves (AIRS data, ERA-I simulation) are almost identical! The black curve is (AIRS minus ERA_simulation), ie blue minus red curve, and it has essentially NO noise because the ERA-I simulations "follow" the AIRS observations very very closely. The droop in the black curve is just the effect of the global increases in $CO_2$ during this time period. Note that in the ERA-I simulations the $CO_2$ amount is fixed at 385 ppm.

**2.5 (5)**

Line 168: "Note that since the ERA-I tracks the atmospheric state quite accurately most of the time-series "noise" is removed" Could the authors provide a reference here? How accurate is ERA-I compared to other models? Here, one should also remember that the authors specifically selected those scenes where ERA-I simulated BT spectral yielded a low residual. I feel that this simply demonstrates that their sampling strategy produced the desired results, not that ERA-I is accurate per se.

This is covered a bit in the previous comment. I cannot provide a reference, this is original work. As stated earlier we did NOT specifically select scenes where ERA-I simulated BT spectrall yielded a low residual, we showed that this selection ONLY used a window channel (not a channel high the atmosphere like this one) and that we only selected out marine boundary layer stratus decks.

More importantly, we **do not use** the ERA-I simulations in this work, they are shown here just for context, and to make the case that ERA-I model fields **are** sufficiently accurate for computing the Jacobians used to do the retrievals. Please see new material in Sec. 4.2 and an extensive evaluation in Table 4 on the effect of any inaccuracies in ERA-I for Jacobian retrievals. They are so small they can be ignored.

**2.6 (6)**

Line 202: "These a-priori covariance uncertainty terms improved simulated retrievals and profile trends generated from these retrievals by 3-10%." Could the authors elaborate on this result? It appears like a large range and I'm wondering if improvements were limited to specific latitudinal zones or regions.

Quite the opposite. Almost all infrared retrievals, especially those trying to measure $CO_2$, use fairly strict a-priori uncertainties in order to regularize the retrieval solutions. Here we use almost no a-priori constraints, and rely almost completely on empirical smoothing regularization ($L$ in Eq. 5) rather than a-priori constraints. We did find that some very loose a-priori constraints helped a little, which is the quote above. 3-10% improvement in the retrievals refers to improvements in the **anomaly** retrievals of temperature and humidity. (1) Neither of those quantities are used to determine AIRS stability, (2) The range of temperature in the anomaly retrievals is ~3K max, so 10% of 3K is 0.3K, not much, and (3) similarly for water vapor, these are very small shifts.

So, it's really quite amazing that these retrievals can be done with very little a-priori constraints, which is most likely due to the fact that anomaly retrievals are insensitive to both AIRS calibration biases and RTA biases!

The effect of these a-priori constraints were very similar with latitude.

**2.7 (7)**

Section 4.3 (Lines 250-259): This section is confusing to me. Could the authors better explain Figure 7? Is the panel on the left, "-55 deg latitude CO2 retrieval" for a single scene?

Section 4.3 is now Section 4.4.

There was a mistake in labelling in Fig. 7, right panel, that could have lead to some confusion, now fixed. As stated in the paper, all anomaly retrievals were done using 16-day clear scene averages binned into 40 equal area latitude bins. There are no "single scene" retrievals in this paper, and no retrievals using absolute radiances (or BT spectra). This is stated clearly in Equations 2 and 3 where our observable $y$ is defined.

This figure addresses how we accounted for the retrievals mixing up temperature and $CO_2$ in the retrieval process. This is a common problem, we now have included some context on this from a EOS AURA-TES paper. Overall we have devoted about 1+1/2 pages to this approach. It's new and our results show tremendous self-consistency. Given that this is a new novel approach, most researchers stumble on it. However, in order to remove doubts, we have added a new row (number two) to Table 3, which summarizes our $CO_2$ trend results. In this new entry we show the measured drifts of our AIRS retrieved $CO_2$ anomalies relative to the ESRL in-situ $CO_2$ anomalies **without** using our correction that accounts for the co-linearity of the $CO_2$ and temperature Jacobians. The drift relative to ESRL is still very small (even smaller than what we deem our correct result) but it does have almost 3X higher statistical uncertainty.

We hope that the expanded discussion of this in Section 4.4 helps the reviewer understand the approach better.

**2.8 (8)**

Figure 9: How is it that the AIRS-ERA SST trend is a perfectly straight line across all wavenumbers?

This figure is showing the fitted linear trend in the time series of the anomaly residuals. The shortwave region residuals have a drift in them, something that could not be fitted out over time using the longwave and midwave channels. We conclude that the short wave is drifting. The (AIRS - ERA) SST trend line is the difference in trend (K/year) between the AIRS retrieved trend (K/year) in SST (from the anomaly retrievals) and the trend in the ERA-I SST product. There is only one trend for either product! What this line shows is that the longwave and midwave AIRS channels produced a SST trend that is far closer to zero than the un-fit trend in the AIRS shortwave channels.

**2.9 (9)**

Lines 567-568: "This work emphasizes that users of AIRS data for climate applications must pay careful attention to channel selection since certain detector arrays and channels are presently not suitable for climate trending, including all of the AIRS short wave channels"

- By "AIRS data", do the authors mean L1c?

Changed to say both L1b and L1c.

- The authors demonstrated that they could calculate the shortwave spectral drift after the fact and, when subtracting it from the retrieved trend in Susskind et al. (2019), they could correct the trend sufficiently. Would such an approach not be a suitable alternative to channel selection? I imagine that the range of geophysical retrievals possible from bias-free channels must be reduced. This gives rise to the question whether climate-quality retrievals could be made from a reduction in spectral channels.

Not sure I understand. You always need a variety of channels since the geophysical trends will always be mixed together in may channels. The AIRS Level 2 algorithm has bias-correction estimates applied to the BT radiance spectra, who knows what that is doing. I spoke with the AIRS Level 2 implementers on this (John Blaisdell) and they found that they could not retrieve emissivity and surface temperature together using only the longwave window channels. Therefore, they used just the shortwave to get surface temperature, and did not vary that when they retrieved surface emissivity (and some of the water column) using only the longwave channels. So, it's a mess in terms of climate.

But, yes, if biases are removed, the retrieval should be more accurate. But, I think the approach used here if far better, just retrieve the anomaly trends, since they dont' contain AIRS calibration biases or RTA biases. Generally the climate community is primarily only interested in anomalies.

- How do the authors envisage the practical implementation of their recommendation here? The method the authors present here appears nuanced and expensive, not easy to implement by users of AIRS data.

Agreed. We have not yet explored how much scene dependence there might be on the offsets in the radiances caused by the AIRS events. I believe it needs to be done by the AIRS/SNPP Projects.

- Do the authors consider publishing a list of AIRS channels suitable for climate applications?

We plan to add the channel list to a repo that goes with this article.

**3 TECHNICAL ISSUES:**

- Discussion of spectral features: It will help the reader a great deal if the authors specify the wavenumber range they refer to with each mention of specific features, e.g., "CO2 region" in Line 160, or "upper-tropospheric water vapor" in Line 162, or "window region...water bands" in Line 207, etc.

Done.

- Line 28: "sea surface temperatures." Define the acronym "SST" upon first use.

Done.

- Line 59: "After a summarizing"

Done.

- Line 89: "Radiance (BT) anomalies" this is confusing since "BT" is an acronym or Brightness temperature, not Radiance.

Tried to clarify. But most people are conversant with interchanging radiance with BT.

- Line 106: "are matched to each clear scene are also saved"

Done

- Figure 1 caption: "Density of AIRS clear ocean scene for calendar years 2012" should be "Density of AIRS clear ocean scenes for calendar year 2012"

Done

- Figure 2: Caption: Should it not be "long-wave" instead of "short-wave"? Since the authors make specific reference to a CO2 feature, would they consider expanding the x-axis and adding more detailed tick marks to help the reader identify this feature specifically? As reader, I have the same issue with Figure 3 and its subsequent discussion.

Yes, fixed, thanks. This figure was substantially changed in response to this reviewer, as discussed above.

- Y-axis label: Reference to B(T) instead of BT. (Same in Fig. 4, Fig. 9)

Fixed
**All the rest of the comments below have been addressed**.

- Line 161: "by by"
- Line 203: "each observations"
- Lines 202-203: Awkward sentence. Meaning unclear.
- Line 225: Add a comma to ease reading: "As discussed in Section 2, only"
- Lines 238, 264, 516: "(Aumann)" reference needs a date.
- Line 254: "to to"
- Figure 6 (page 11): legend should probably be "All channels used" for the blue profile?
- Line 267: "every so slightly"
- Line 294: "channels in located below 1615 cm-1"
- Line 300: Could the authors provide a reference for the L2c product here, so that the reader could follow up and better understand how "channels that do not exist . . .are filled during L1c creation". Printer-friendly version
- Line 312: "discussed in Sect. sec:sst"
- Line 316: "results presented here use avoid the short wave"
- Line 418: "just two two small"
- Line 546: "improvements to the AIRS products can be improved"
- Line 564: "jumps are observed in the all retrieved" Interactive comment on Atmos. Meas. Tech. Discuss., doi:10.5194/amt-2019-504, 2020.

[revised manuscript text omitted]

---

## Author Response (AR2)

**Comments to the Editor**

Associate Editor Decision: Publish subject to technical corrections (15 Jul 2020) by Thomas Wagner

**Establishment of AIRS Climate-Level Radiometric Stability using Radiance Anomaly Retrievals of Minor Gases and SST**

by L. Larrabee Strow and Sergio DeSouza-Machado

**Reviewer #1 Comments**

1. Line 30: 'major limitation is that is ...' change 'is' to 'it'
2. Citation format error in Line 31, 619, 674, 676, and 677. (Line number in 'response to reviewer').
3. If possible, consider directly showing trends that pass the statistical significance test, including in Fig.3, Fig.19 + Fig.20, mainly to condense the context you aim to deliver.
4. Can you add an additional panel/curve in Fig.18 to contrast the BT anomaly contribution by retrieved quantities and BT anomalous residual (similar to Fig. 21)?

**Author Replies**

**Request 1.**

Done

**Request 2.**

I do not see any problems with these citations. Hopefully the copy editor might see what I am missing?

**Request 3.**

**Figure 3**

I am very reluctant to highlight this issue because (a) our observation subset is a highly spatially non-uniform set of clear scenes that make general geophysical insights difficult, and (b) we have already shown that our anomaly retrievals are extremely accurate for the minor gas forcings and agree reasonably well with ERA-I for temperature trends (see Appendix). In this work we rely on 95% (2-$\sigma$) uncertainties (often corrected for serial correlation in time series).

The following sentence has been added to partially respond to the Reviewer's suggestion for Fig. 3.

*"Spectral regions in Fig. 3 that exhibit trends smaller than the 2-$\sigma$ uncertainty are often channels where the BT trends that are due to increasing $CO_2$, $CH_4$, and $N_2O$ are counter-balanced by changes of the opposite sign due to trends in either the atmospheric or surface temperature. These counter-balanced trends are all properly accounted for in the anomaly retrievals given the good agreement between the observed and in-situ minor gas trends."*

**Figures 19 and 20**

Figure 19 shows the the mean BT anomalies for the 16-year period broken out by the geophysical contributions to these anomalies derived from our retrievals, as requested by Reviewer #1. In this paper, the $CO_2$ retrieval was the main in-situ data record used to determine the AIRS stability. The statistical uncertainties in these $CO_2$ anomaly retrievals are best determined by comparing to the truth data, which is done in Fig. 10 showing the AIRS-ESRL $CO_2$ anomaly differences, which are discussed in detail in the text.

As I understand it, the reviewer is asking for statistical uncertainties in retrieved parameters as derived from the optimal estimation (OE) process, which is very tricky and not all that relevant to this paper. Since we used a diagonal measurement uncertainty covariance estimate, any parameter retrieval uncertainty estimates will likely be unrealistically small, since the AIRS radiance (noise) can have significant correlations. However, none of the major (NASA and NOAA) systems that do AIRS retrievals include non-diagonal noise covariance. We also expect that the forward model (Jacobian) uncertainties have off-diagonal components as well, which are not included. Therefore we think it best to not estimate something that will likely be incorrect.

**Request 4**

Our answer here is similar to that for Request 3, Figure 3. However, we have added the following sentences in Section 6.1 to help remind the reader that the anomaly residual errors are far below the observed anomalies shown in Fig. 19.

*"The uncertainties in the mean spectral BT anomalies shown in in Fig. 19 can be estimated from the mean differences between the observed and computed BT anomalies shown in Fig. 8. An average over all fitted channels gives a mean residual of -0.0021K ± 0.03K. This excellent fit, combined with the good agreement between the observed and in-situ truth data for the $CO_2$, $CH_4$, and $N_2O$ anomalies indicates that the anomalies shown in the top panel of Fig. 19 are likely accurate to the anomaly fit 2-$\sigma$ uncertainty of level 0.03K."*

Similarly to Request 3, the reviewer is essentially asking us to break out the anomaly fit residuals versus time for each fitted variable and estimate their uncertainties. We could do that, but the point of this paper is to compare to "truth" data, ie $CO_2$, from which we obtain our stability estimate for AIRS. To do this we would take the uncertainties for each fitted variable returned by the OE retrieval, and then multiply that uncertainty by the Jacobian of the geophysical variable. We would prefer to not do that, since we have not really probed the accuracy of our OE uncertainty estimates, which are very sensitive to correlations in the measurement error covariances. Estimating off-diagonal measurement error covariances is very difficult, so for this work we used a diagonal measurement error covariance matrix, which will result in unrealistically low parameter error estimates. We believe the true $CO_2$ measurement uncertainties are best derived by comparing our retrievals to the ESRL $CO_2$ in-situ truth data. This is a common problem in OE retrievals, and we would rather not address it here since it does not impact the main conclusions of the paper, which are retrieval comparisons to truth.

Also, a few, mostly numerical changes, were made in the text regarding the newly inserted Figs. 19 and 20, which can be seen in lines 700-701 and lines 709-710 in the diff.pdf file. We inadvertently used slightly incorrect versions (scaling errors) of these two figures when summarizing them in text, requiring small changes to the discussion from lines 705-710. These changes only impact the descriptive summarizes of the mean anomalies.